# Simulations of the 2005, 1910 and 1876 Vb cyclones over the Alps – Sensitivity to model physics and cyclonic moisture flux

Peter Stucki[1,2], Paul Froidevaux[2,4], Marcelo Zamuriano[1,2], Francesco Alessandro Isotta[4], Martina Messmer[1,3,5], Andrey Martynov[1,2]

[1]Oeschger Centre for Climate Change Research, University of Bern, Bern, 3012, Switzerland
[2]Institute of Geography, University of Bern, Bern, 3012, Switzerland
[3]Climate and Environmental Physics, Physics Institute, University of Bern, Bern, 3012, Switzerland
[4]Federal Office for Meteorology and Climatology MeteoSwiss, Zurich-Airport, 8058, Switzerland
[5]now at: School of Earth Sciences, The University of Melbourne, Melbourne, Victoria, Australia

*Correspondence to*: Peter Stucki (peter.stucki@giub.unibe.ch)

**Abstract.** In June 1876, June 1910 and August 2005, northern Switzerland was severely impacted by heavy
precipitation and extreme floods. Although occurring in three different centuries, all three events featured very similar precipitation patterns and an extra-tropical storm following a cyclonic, so-called Vb trajectory around the Alps. Going back in time from the recent to the historical cases, we explore the potential of dynamical downscaling a global reanalysis product from a grid size of 220 km to 3 km. We use the full, 56-member ensemble provided in the reanalysis and a regional weather model to investigate sensitivities of the simulated precipitation amounts to a
set of differing model configurations. The best-performing model configuration in the evaluation, featuring a 1-day initialization period, is then applied to assess the sensitivity of simulated precipitation totals to cyclonic moisture flux along the downscaling steps. The analyses show that cyclone fields and tracks are well defined in the reanalysis ensemble for the 2005 and 1910 cases, while deviations increase for the 1876 case. In the downscaled ensemble, the accuracy of simulated precipitation totals is closely linked to the exact trajectory and stalling position
of the cyclone, with slight shifts producing erroneous precipitation, e.g., due to a break-up of the vortex if simulated too close to the Alpine topography. Simulated precipitation totals only reach the observed ones if the simulation includes continuous moisture fluxes of >200 kg m$^{-1}$ s$^{-1}$ from northerly directions, and high contributions of (embedded) convection. Misplacements of the vortex and concurrent uncertainties in simulating convection, in particular for the 1876 case, point to limitations of downscaling from coarse input for such complex weather
situations and for the more distant past. On the upside, single (contrasting) members of the historical cases are well capable of illustrating variants of Vb cyclone dynamics and features along the downscaling steps.

## 1 Introduction

Floods are among the most damaging natural hazards worldwide (Bevere et al., 2018); they affect more people than any other natural hazard (CRED 2019). The costliest flood event in Switzerland of the last decades occurred in
2005 (Hilker et al. 2009); it caused fatalities and led to heavily damaged infrastructure (Bezzola et al. 2008). This event was well documented and subsequently, a range of publications analyzed the flood-inducing meteorological conditions (e.g., Frei 2005; Beniston 2006; MeteoSwiss 2006; Bezzola and Hegg 2007; Zängl 2007a; Bezzola and Hegg 2008; Hohenegger et al. 2008; Jaun et al. 2008; Langhans et al. 2011; Stucki et al. 2012; Messmer et al. 2017).

On a synoptic scale, the associated extratropical cyclone mainly followed the classical so-called Vb cyclone track after Köppen (1881) and van Bebber (1891), see also Hofstätter et al. (2016). Cyclones on a Vb track are associated with heavy to extreme precipitation over Central Europe, and particularly north of the Alps (Hofstätter et al., 2016, 2018; Nissen et al., 2013). Most of the cyclones following the Vb trajectory are generated in the western

Mediterranean region, and most of them pass the Golf-of-Genoa region (Hofstätter and Blöschl, 2019; Messmer et al., 2015). For this, they are also called Genoa Lows at this stage; Bezzola et al. 2008). They take up moisture over the Ligurian Sea, then propagate eastward to the Adriatic Sea and recurve northward (Hofstätter et al., 2018; Hofstätter and Blöschl, 2019; Messmer et al., 2015; Pfahl, 2014; Ulbrich et al., 2003). Regarding the pattern of mid-tropospheric geopotential heights, the 2005 case has also been characterized as a pivoting cut-off low (PCO;

Stucki et al. 2012; Froidevaux and Martius 2016) referring to the recurving track of the system around the Alps while its axis of symmetry turns from meridional to zonal; see also Awan and Formayer (2017) for a general description of cut-off lows and their influence on extreme precipitation in the European Alps. Furthermore, quasi-stationarity (i.e., stalling) of the system over northern Italy was important: In the cyclonic circulation around the Vb cyclone, large quantities of warm and humid Mediterranean air were led over and around the Eastern Alps. This

dynamic mechanism is described as "cyclonic moisture flux" hereafter. With the term "cyclonic", we refer to the pattern of the moisture flux streamlines, forming a (closed) anticlockwise rotation around the cyclone center. Downstream, the cyclonic moisture flux impinged onto the slopes of north-eastern Switzerland from sector North (Hohenegger et al. 2008; Froidevaux and Martius 2016; Messmer et al. 2017). The intensity of the integrated water vapor transport (IVT) was estimated to exceed 300 kg m$^{-1}$ s$^{-1}$ (Froidevaux and Martius 2016). Hence, IVT was

found to be an important precursor for severe floods in Switzerland (cf. Kelemen et al., 2016, for a European summer flood in 2013). Precipitation occurred in two peak episodes in the afternoons of 21 and 22 August 2005, respectively, when stratiform upslope orographic precipitation was locally enhanced by embedded convection (Hohenegger et al. 2008; Langhans et al. 2011).

Although the impact of the 2005 event was very severe, it was not unique in a historical context. Several studies

found similar spatial distributions of damage and precipitation, as well as similar synoptic-scale weather patterns. Two such analog cases occurred in June 1910 and June 1876 (Röthlisberger 1991; Pfister 1999; Frei 2005; Stucki et al. 2012). For instance, their similarities were analyzed on a synoptic scale using the Twentieth Century reanalysis (20CR; Compo et al. 2011), and classified as PCO type 1 (2005, 1910) and type 2 (1876), where the 1876 case features a more north-westerly flow towards the Swiss Alps (Stucki et al. 2012).

However, two options for hydro-meteorological analyses have not been considered so far to learn from these historical cases. The first option is the systematic use of a reanalysis ensemble to assess sensitivities of the severe weather with regards to determining factors such as cyclone trajectories or IVT. The second option is using the global reanalysis products for dynamical downscaling to meso-scale resolutions, i.e. the nesting of limited-area weather models into larger-scale models in several refinement steps (von Storch et al., 2000; von Storch and Zorita,

2019). In fact, 20CR has proven to be a valuable input dataset for downscaling heavy-precipitation and windstorm events over the Central Alps back to the 19th century. Stucki et al. (2018) showed that downscaling the ensemble mean is not only computationally cheaper, but can be seen as a minimum-error and thus natural approach in well-represented areas and distinctive synoptic flow conditions. For an extreme flood in 1868, they found a small smoothing effect of the associated cyclone, which induced southerly moisture flux, i.e. perpendicular to the Alpine

range. In contrast, Hohenegger et al. (2008) used a limited-area ensemble prediction system to assess potential benefits for precipitation forecasting. They found that member-to-member variability tends to have a larger effect than resolution. Other studies point to limited benefit from the full ensemble and recommend using a set of well-chosen members (Jaun et al. 2008; see also Horton and Brönnimann, 2018, for statistical downscaling of precipitation fields).

For dynamical downscaling, there are manifold options regarding the configuration of the limited-area weather models, including the choice of adequate reanalysis products as input datasets, initialization spans (so called spin-up), spatial extent and resolution of the simulation domains, or model physics (e.g., Prein et al., 2015). To date, only 20CR covers the 1876 case, at the expense of a coarse grid size of 2° x 2° in the horizontal, while finer-resolved reanalysis products have been used to downscale cases after 1900 (e.g. Brugnara et al. 2017). Regarding the initialization, long spin-ups would allow soil moisture and similarly slow-adapting model variables to reach an equilibrium, at the expense of potentially losing control over the simulation with time. In turn, short spin-up times constrain the potential evolution close to the large-scale input. For Vb cyclones, relatively short spin-ups were chosen by Hohenegger et al. (2008) and Messmer et al. (2015, 2017). Regarding spatial resolution, cloud-resolving and convection-permitting grid sizes equal to or lower than about 3 km are necessary to reproduce the precipitation of the 2005 case (Zängl 2007b; cf. Prein et al., 2015). Typical setups of limited-area models mostly include explicit production of precipitation in the innermost domain, while convection is parametrized for the coarser domains. Further options are one- versus two-way nesting or nudging in one or more domains.

In this study, we assess the 2005, 1910 and 1876 cases in two ways. First, we aim to find a setup of the Weather Research and Forecasting (WRF) model (Skamarock et al., 2008) that is adequate for dynamical downscaling from 20CR and for our cases. For this, we use the 2005 case as testbed because of the large amount of observations available for verification. Second, we apply the chosen setup to all three cases, aiming to investigate relevant atmospheric features that induce heavy precipitation, and to assess the inevitably increasing uncertainty along the downscaling steps and among the ensemble members as we go back in time from 2005 to 1910 to 1876.

The article is organized as follows. Data and models are introduced in Section 2. The experiments with different model setups are described in Section 3. The synoptic and meso-scale reconstructions of the three cases are presented in Section 4. A summary and conclusive remarks are given in Section 5.

## 2 Data and Models

### 2.1 Observation-based precipitation datasets

All observation-based precipitation datasets used in this study come from the Federal Office for Meteorology and Climatology MeteoSwiss. Precipitation totals derived from all three observation-based products are shown in Figure A1 in the Appendix. The first dataset are observations of daily precipitation totals. These measurements are quality checked and homogenized according to Füllemann et al. (2011).

CombiPrecip, the second product, is also a gridded dataset. It results from a geo-statistical combination of rain-gauge observations and radar images. It covers the entire Swiss territory for the period 2005 to present at high spatial and temporal resolutions. The hourly precipitation accumulation is available as a running sum updated every 10 minutes on a 1-km grid.

The third MeteoSwiss product is RrecabsD, a prototype dataset specifically calculated for this study with a statistical reconstruction technique. The procedure was previously used to reconstruct monthly and daily precipitation in the Alps for different scopes (Isotta et al., 2019; Masson and Frei, 2016; Schiemann et al., 2010; Schmidli et al., 2002). It involves a Principal Component Analysis (PCA) of a high-resolution grid dataset in a calibration period defined between 1981 and 2010 and an Optimal Interpolation of PCA scores from long-term station data. The high-resolution dataset used for the PCA is RhiresD (MeteoSwiss, 2013) on a grid-size of 2.2 km, with daily precipitation totals retrieved by spatial interpolation of rain-gauge measurements within the Swiss borders. In RrecabsD, the focus is on spatial consistency by using all station measurements available both in the respective days in 1910 or 1876 and in a consistent part of the calibration period (1981 - 2010), which is accordingly slightly reduced by eliminating the days with gaps at one or more of the chosen stations. The three datasets described above integrate observational information and are used as reference. They are, like other dataset, affected by uncertainties and errors, which were analyzed in detail in the provided references and in several applications.

## 2.2 Reanalysis datasets

The Twentieth Century Reanalysis dataset version 2c (20CR; Compo et al., 2011) is used for synoptic analyses and as initial and boundary conditions for the downscaling experiments. 20CR is a global atmospheric reanalysis with a 2-degree spatial grid (approx. 220 km over Europe), 28 vertical hybrid-sigma pressure levels and 6-hourly temporal resolution going back to 1851. Only surface pressure observations are assimilated. Over the time period of the 1876 case, a number of four stations within Switzerland are used in the assimilation (Bern, Sion, Grand St. Bernard, and Geneva). This number grows to six for the 1910 case (including also Zurich and Basel), and to 34 for the 2005 case. The 20CR ensemble mean and 56 members (in fact, 1$^{st}$-step deviations from the ensemble mean) are available.

The ERA-Interim (Dee et al. 2011) and CERA-20C reanalyses (Laloyaux et al. 2016) are used for comparisons of synoptic fields to 20CR. ERA-Interim (CERA-20C) has a horizontal grid size of approx. 80 km (125 km), 37 (37) pressure levels, 6-hour (3-hour) temporal steps and reaches back to 1979 (1901). ERA-Interim is also used as initial and boundary conditions for downscaling the 2005 case to compare with downscaling based on 20CR.

## 2.3 Regional model WRF

The non-hydrostatic Weather Research and Forecasting model (WRF-ARW Version 3.7.1; Skamarock et al. 2008) is used for dynamical downscaling of the 2005, 1910 and 1876 cases. An initial model setup is applied to downscale the 2005 case, and a number of nine follow-up setups are used to investigate differences among the setups regarding the representation of precipitation totals, and to select a best setup to downscale the 1910 and the 1876 cases.

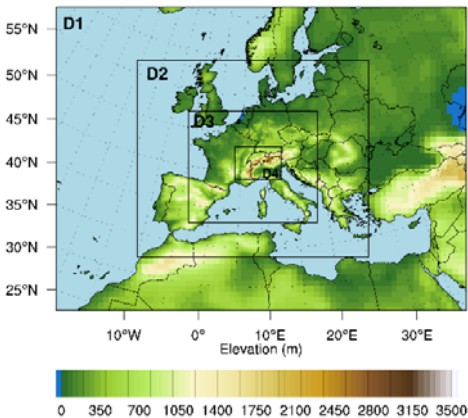

**Figure 1: Nesting of the WRF regional model into 20CR, where D1 to D4 refer to the simulation domains with 81-km, 27-km, 9-km and 3-km grid sizes. Shading indicates model topography for each domain in m a.s.l.**

Here, we describe the initial, standard setup, which is used to downscale each of the 56 ensemble members in 20CR. Model initialization is set to 11 August 2005, while the largest precipitation totals were observed on 22 August 2005. This is based on the original assumption that an ample spin-up time of more than a week is desirable for the inner domains to reach some internal equilibrium. For example, the accumulation of soil moisture can significantly contribute to enhanced convection and precipitation, and should therefore be considered (Zbinden,
2005; MeteoSwiss, 2006; Bezzola and Hegg, 2007; Cioni and Hohenegger, 2019). The horizontal setup consists of four nested domains with grid sizes of 81, 27, 9 and 3 km. The innermost domain covers much of the Alpine bow to avoid complex terrain at the boundaries (Figure 1). The vertical setup consists of 60 eta levels (with a top level of 50 hPa) to capture fine-scale features of vertical lifting and condensation, and the Thompson microphysics scheme (Thompson et al. 2008) is used for bulk microphysical parameterization. Additionally, we use the Yonsei
University (YSU) planetary boundary layer (PBL) scheme (Hong et al., 2006) for resolving turbulent fluxes with a complex orography effects' correction to the finest domain (Jiménez et al., 2012). The Kain Fritsch cumulus parameterization is used in the larger domains (Kain, 2004), and turned off in the innermost domain. Moderate spectral nudging (corresponding to a wavelength of about 1500 km) is applied to temperature, wind and geopotential fields above the PBL in the outermost domain to ensure consistency with the large scale forcing (von
Storch et al., 2000). The downscaling output is stored in hourly resolution.

**3 Downscaling of the 2005 case**

**3.1 Downscaling with one initial and nine modified setups**

All 56 ensemble members of 20CR are downscaled with a standard WRF setup (see Sect. 2) in the first step. This is done for the 2005 event, as only for this case, simulations can be verified using a state-of-the-art spatial
reconstruction of precipitation totals, which is CombiPrecip in our case. For the verification, we focus on a control area in north-eastern Switzerland (see the small box in Figure A1 in the Appendix), the region where most of the precipitation fell, and on precipitation totals over 48 hours starting from 21 August 2005 06 UTC, i.e. the two-day period with highest precipitation intensities (MeteoSwiss, 2006). In fact, precipitation was concentrated over north-eastern Switzerland in the 2005 case, with gradients from the Alpine crests towards the Swiss Plateau. This spatial

distribution was also important for the subsequent flooding (Bezzola and Hegg, 2005). Results show a general underestimation of the accumulated precipitation in the control area (Figure 2). Furthermore, CombiPrecip shows high precipitation rates over most of the analyzed period. In contrast, the downscaled ensemble has only two periods of very high precipitation rates (at around 12 and 36 hours after initialization), and obviously, there is too little

precipitation between these two simulated peak episodes. In addition, the spread of precipitation totals in the ensemble is large. For instance, the median member (#15) underestimates the mean accumulated precipitation in Northern Switzerland by a factor of four: approx. 20 mm/48 h versus 80mm/48 h in CombiPrecip. Only one member (#24) reaches around 50 mm/48 h, while the lowest member (#23) produces only around 10 mm/48 h.

In all, we cannot be satisfied with these results yet. On the one hand, we aim to investigate particularly flood-

inducing features of Vb-cyclones. For this, a certain variability in the ensemble is helpful and necessary. For example, we can find and assess (non-)decisive features by means of opposing ensemble members. On the other hand, we also need to ensure that our downscaling experiment delivers plausible results, especially regarding precipitation intensities and patterns. For this, the deviations from the observations must not become too large in the ensemble. A somewhat smaller spread of the simulated precipitation for the 2005 case would also increase our

confidence that the simulation of historical events will produce reasonable and valid results.

In short, we would have expected less underestimation and smaller deviations with this downscaling configuration (cf. Coppola et al., 2018, their figure 5, for estimations of accumulated precipitation over the Alps from a multi-model ensemble). One possible reason for the large variability is the very long spin-up time of 10 days, which may let the simulations run too freely, i.e. independently from the synoptic reanalysis data.

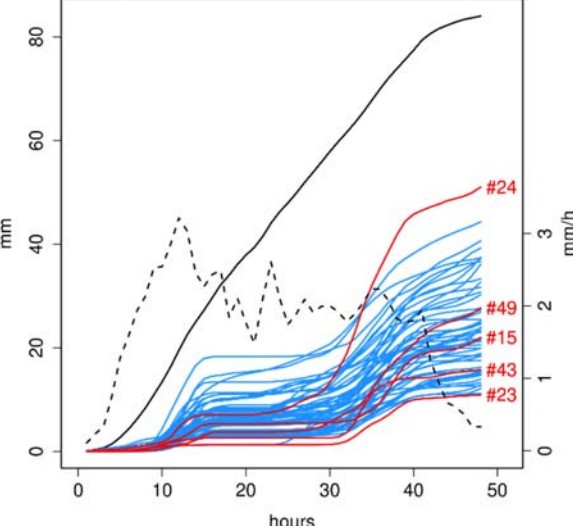

**Figure 2: Cumulative totals of hourly precipitation (mm, left y-axis) over 48 hours starting at 2005-08-21 07 UTC (x-axis), as simulated in 56 members (blue lines) that are downscaled from 20CR. Mean values are given for a control area in north-eastern Switzerland. Ensemble members (#23, #43, #15, #49, #24) representing the quartiles of the distribution**

**are highlighted in red. For comparison, cumulative totals in the CombiPrecip dataset is shown (black line), as well as the hourly time series in CombiPrecip (dashed line; mm on the right y-axis).**

Accordingly, a second set of experiments is done with decreasing spin-up periods from 10 to 7, 5, 3, and 1 days (see also Table 1 and Table 2 for details). With a 1-day spin-up, both early and late onsets of the most intense precipitation in different regions of the Central Alps are still captured. To save computational costs, these tests are done with a subset of ten members that cover the full range of precipitation variability from the original setup (not

shown). In terms of precipitation over northern Switzerland, the best results are achieved with a spin-up time of 1 day (see Sect. 3.2 below for details of the full evaluation).

**Table 1: Abbreviations and description of setup experiments.\***

| NAME | INITIALIZATION | OTHER MODIFICATIONS |
|---|---|---|
| sp10 | 2005-08-11 00 UTC | initial setup, see Sect. 2 |
| sp7 | 2005-08-14 00 UTC | |
| sp5 | 2005-08-17 00 UTC | |
| sp3 | 2005-08-19 00 UTC | |
| sp1 | 2005-08-20 06 UTC | |
| sp1_dom | 2005-08-20 06 UTC | larger domains, grid ratio=5 |
| sp1_cu | 2005-08-20 06 UTC | cu_physics=11, |
| sp1_cu_mp | 2005-08-20 06 UTC | cu_physics=11, mp_physics=95 |
| sp1_cu_nudg | 2005-08-20 06 UTC | cu_physics=11, nudging also 2nd domain |
| sp1_cu_nest | 2005-08-20 06 UTC | cu_physics=11, two-way nesting |

**\*Changes are indicated with respect to the original setup with a 10-day spin-up period (sp10): The abbreviation sp5 indicates a change in the spin-up period to 5 days, mp indicates a change of the microphysics scheme, cu a different cumulus scheme, nudg indicates nudging in two domains, nest indicates two-way nesting, and dom stands for a larger**

**domain. See text for details.**

In a third set of experiments, potential enhancements are explored by applying further modifications to this last setup. The goal of these last experiments is not to achieve a robust sensitivity assessment for each tuning option, but to determine a better setup for our purposes by further modifying a number of configurations of the WRF

model.  Concretely, we test larger domains since the high-resolution domain does not cover the entire extent of the cyclone. For this test, we also increase the grid ratio from three to five. Another test involves changing the cumulus scheme from Kain Fritsch to Multi-scale Kain Fritsch (Kain, 2004). Multi-scale Kain Fritsch contains a grid resolution dependence, which may improve the location and intensity of precipitation in high-resolution simulations (Zheng et al., 2016). Then, the microphysics scheme is changed from Thompson et al. (2008) to Ferrier

et al. (1995), where, among other differences concerning hydrometeors, ice, graupel and hail are described in less detail. Furthermore, moderate nudging in the two outermost domains is applied to test for the effect of keeping the simulation close to initial conditions. Finally, one-way is compared to two-way nesting, allowing also for exchange of information from finer to coarser domains.

## 3.2 Evaluation of ten WRF model setups for precipitation totals

We use three evaluation methods to determine an overall best-performing model setup for downscaling our cases. Again, we focus on the control area in north-eastern Switzerland and on precipitation totals over 24 and 48 hours starting from 20 August 2005 06 UTC. This is because we are particularly interested in finding a setup that would ideally produce correct precipitation maxima in the correct areas on the regional and local scales. CombiPrecip is again used as the reference, i.e. observation-based dataset (see above and Sect. 2). Recall that CombiPrecip also includes radar information. Both CombiPrecip and the WRF simulation are bilinearly interpolated to a 6-km horizontal grid for comparability and to reduce single-cell effects.

**Table 2: Evaluation of WRF setups for dynamical downscaling.***

| VER | sp10 | sp7 | sp7 _mp | sp5 | sp3 | sp1 | sp1 _dm | sp1 _cu | sp1 _cu_nd | sp1 _cu_ns |
|---|---|---|---|---|---|---|---|---|---|---|
| MAE 24h | 14.50 | 14.87 | 14.96 | 14.27 | 16.90 | 13.38 | 13.30 | 16.14 | 15.32 | 16.22 |
| MAE 48h | 27.75 | 28.69 | 28.70 | 26.89 | 25.70 | 25.44 | 27.45 | 24.45 | 28.79 | 26.99 |
| BOX 24h | 0.90 | 0.90 | 0.74 | 0.86 | 1.00 | 0.64 | 0.70 | 0.94 | 1.37 | 1.05 |
| BOX 48h | 0.59 | 0.58 | 0.46 | 0.54 | 0.67 | 0.43 | 0.43 | 0.62 | 1.05 | 0.73 |
| **Rank** | 5 | 5 | 4 | 3 | 8 | 1 | 2 | 5 | 10 | 9 |

| VIS | sp10 | sp7 | sp7 _mp | sp5 | sp3 | sp1 | sp1 _dm | sp1 _cu | sp1 _cu_nd | sp1 _cu_ns |
|---|---|---|---|---|---|---|---|---|---|---|
| 24 h | 10 | 8 | 10 | 13 | 7 | 14 | 13 | 8 | 1 | 4 |
| 48 h | 10 | 9 | 13 | 12 | 4 | 15 | 13 | 8 | 1 | 4 |
| **Rank** | 5 | 6 | 4 | 3 | 8 | 1 | 2 | 7 | 10 | 9 |

| EMD | sp10 | sp7 | sp7 _mp | sp5 | sp3 | sp1 | sp1 _dm | sp1 _cu | sp1 _cu_nd | sp1 _cu_ns |
|---|---|---|---|---|---|---|---|---|---|---|
| 24 h | 0,231 | 0,279 | 0,282 | 0,214 | 0,273 | 0,117 | 0,135 | 0,11 | 0,146 | 0,13 |
| 48 h | 0,159 | 0,171 | 0,209 | 0,13 | 0,116 | 0,096 | 0,12 | 0,099 | 0,12 | 0,12 |
| **Rank** | 8 | 9 | 10 | 7 | 5 | 1 | 4 | 1 | 5 | 3 |

**\*Spatial verification measures for evaluating the performance of ten different WRF simulation setups (columns; see Table 1 for abbreviations) against observations (CombiPrecip fields) of 24-h and 48-h precipitation totals in a rectangle box over north-eastern Switzerland (see Figure A1), each calculated over a subset of 10 ensemble members. The top section shows two verification (VER) measures, i.e. (i) mean absolute error (MAE) of simulated versus observed precipitation totals (mm/24 h or mm/48 h), and (ii) mean absolute deviation of the simulated box ratios from the observation-based box ratio (specific rows denoted with BOX). For instance, the value of 0.9 (0.59) for sp10 and 24-hour (48-hour) precipitation totals indicates the mean absolute deviation from the observation-based value of 1.99 (2.26) in CombiPrecip. The middle section shows the scores from visual inspection (VIS), and the bottom section considers the spatial distribution of precipitation inside the box (EMD; see text for details). Ranks are added to all sections; ties are set equal.**

The first evaluation is based on two spatial verification measures, i.e. mean absolute error (MAE) of the simulated versus CombiPrecip precipitation inside the control area, and a metric using box ratios (see section VER in Table 2). The MAE measures the average distance between forecast and observation, and is preferred over RMSE because it is more resistant to outliers, and over correlation coefficients because we are more interested in accuracy than

linear association (Joliffe and Stephenson, 2012). Box ratio means the ratio of mean precipitation in the control area, i.e. the small box w.r.t. a larger box that encompasses Switzerland (see Figure A1 in the Appendix). The box ratio indicates how much of the precipitation is simulated in the correct region when compared to CombiPrecip. The box ratio in CombiPrecip for the 1-day (2-day) mean precipitation is 1.99 (2.26). In simple words, the mean observed rainfall was about twice as intense in the control area over north-eastern Switzerland compared to all of

Switzerland. For the evaluation, we calculate the mean absolute deviation of the simulated box ratios from the box ratio in CombiPrecip.

The second evaluation is based on visual inspection of the simulated precipitation totals, with a focus on the highest amounts of precipitation, that is, the 4th quartile. Again, the according precipitation totals in CombiPrecip are used as a reference. The inter-subjective judgement by the authors yields two points per downscaled ensemble member

for a 'good' match, one point for a 'fair' match, and zero points for 'mismatch' (see section VIS in Table 2). A good match is achieved in case the simulation places the highest quartile of precipitation in the correct regions when compared to the spatial patterns in CombiPrecip.

To contrast the subjective judgements, the third evaluation uses the Earth Mover's Distance (EMD; Rubner et al. 1998, 2000) as a purely objective metric of similarity. The EMD, sometimes known as the Wasserstein distance, is

typically used for pattern recognition in digital image processing, and has as well been applied in atmospheric sciences, e.g., to pollutant concentrations, top-of-atmosphere radiation fluxes, time series of wind maxima, or precipitation and climate indices (Düsterhus and Hense 2012; Baker et al. 2013; Farchi et al. 2016; Düsterhus and Wahl 2018). Intuitively, it measures the cost (mass times distance) of turning one pile of dirt in one area into a second, reference pile with the same overall mass and covering the same area. For our case, this means that the

precipitation fields are normalized, and hence, the EMD considers only the relative patterns of precipitation. Specifically, the EMD indicates how well the simulated spatial distribution of the precipitation matches the observed distribution on a 6-km grid inside the control area. EMD was chosen over typical feature-based methods (e.g., the SAL method by Wernli et al., 2008) because it yields one number, involves less subjective choices and thresholds, and emphasizes the relative pattern. Section EMD in Table 2 gives the median distance for each setup

and for 24-h and 48-h precipitation totals.

The evaluation shows substantial differences in overall performance and ranking of the ten WRF setups. In the first place, Theil-Sen slope estimates are calculated over sp10, sp7, sp5, sp3, and sp1 for all measures; they are all negative (MAE 24 h: -0.2; MAE 48 h: -0.70; BOX 24 h: -0.03; BOX 48 h: -0.03; VIS 24 h: -0.75; VIS 48 h: -1.13; EMD 24 h: -18.5; EMD 48 h: -17.9). Although the trends are not significant in Mann-Kendall tests (or not clearly

attributable, due to the small sample), the negative slopes indicate that performance generally increases with decreasing spin-up time. We infer from this that the Vb cyclone should already be located within the outermost WRF domain at the time of initialization. This allows the WRF model to better track the evolution of the storm system. In the end, the setup with the shortest spin-up is the best ranked. Secondly, the Ferrier and Thompson microphysics schemes perform similarly well. This might be because 2005 is a summer case with rather high

temperatures and hence, the variety in hydrometeors is not very large. Therefore, the Thompson microphysics is selected, which is commonly used in studies on simulating precipitation in complex terrain (e.g., Parodi et al., 2017). Thirdly, further experiments based on a one-day spin-up do not result in better overall performance: neither changing the cumulus scheme, applying two-way nesting, nudging of shorter wavelengths nor using a larger

innermost domain result in a better representation of the precipitation over Northern Switzerland during the 2005 event.

Overall, the best ranked WRF setup in all three evaluations is the standard setup with a one-day spin-up (sp1 in Table 2). In the following, this setup is used for the simulations of the 2005, 1910 and 1876 cases.

For a comparison with better-resolved input data, the selected setup is contrasted to a downscaling experiment with

the same simulation setup and WRF version, but with initial and boundary conditions from the ERA-Interim dataset. The experiment yields an EMD value of 0.2, and 48-hour precipitation totals of up to 350 mm (Figure A2 in the Appendix). While the EMD value in this simulation is in the range of the best downscaled 20CR members, precipitation is much higher. We infer from this that downscaling from 20CR reproduces the relative distribution of precipitation equally well, while the higher intensities may be attributed to the better spatial resolution of

moisture variables in the ERA-Interim reanalysis. In addition, specific humidity at 1000 hPa is higher in 20CR over the Alps, but higher over Eastern Europe in ERA-Interim (not shown). Hence, the advection of moist air to the central Alps is arguably stronger in ERA-Interim.

## 4 Analyses and simulations of the 2005, 1910 and 1876 cases

### 4.1 Precipitation, cyclone fields and tracks in the 20CR ensemble

In this section, we analyze how well the three cases are represented in the 20CR members on a synoptic scale (Figures 3 and 4). For this, we compare data from 20CR to data from ERA-Interim (only available for 2005) and CERA-20C (available for 2005 and 1910). Specifically, we analyze the large-scale patterns of precipitation totals during the most intense phases (21 - 22 August 2005, 13 - 14 June 1910 and 11- 12 June 1876). Furthermore, we investigate the synoptic setting and intensity of the associated cyclone in the ensemble members.

For the analyses, we use both sea level pressure (SLP) and mid-tropospheric pressure fields. SLP fields inform about the quality of the assimilation process in 20CR, and the isobaric pressure fields (at 500 hPa here) tell about the derivation of upper-air variables from the SLP information in 20CR. Combining SLP and isobaric levels has been found useful for cyclone tracking (Hofstätter and Blöschl, 2019). The cyclone tracks shown in Figure 4a, b, and c are reconstructed as follows: In a first step, absolute minima of the 500-hPa geopotential height are

inventoried. Then, the cyclone centers (here, the absolute minima of geopotential height) closest to Corsica on 22 August 2005, 13 June 1910 and 11 June 1876 are selected. In a third step, cyclone tracks are reconstructed every six hours backward and forward in time by selecting the closest cyclone position and starting from the three selected cyclones. The tracks are terminated if the cyclone position jumps over more than approximately 1200 km in six hours. For example, an absolute minima of the 500 hPa geopotential height (here, a cyclone center) exists in 36 of

the 56 ensemble members over southern England on August 20, 00UTC (Figure 4a). One day later, all 56 members contain a cyclone center over southern France. For comparison to the mid-tropospheric level, cyclone fields (Figure 4d, e, and f) and tracks (Figure 4g, h, i) are also calculated for SLP according to Wernli and Schwierz (2006; see

also Welker and Martius, 2015). The algorithm detects cyclone fields in terms of a finite area around a regional SLP minimum, that is, by a closed SLP contour line. The regional SLP minima for each cyclone life cycle are stored as cyclone tracks, and the presence or absence of a cyclone is represented in a binary field for each grid point and time step.

Inferring from Figure A1 in the Appendix, as well as from analyses of supra-national rain gauge measurements (Frei 2006; Stucki et al. 2012) or model simulations (Langhans et al. 2011, for the 2005 case), most precipitation is expected to accumulate over north-eastern Switzerland, and to reach well into Austria and south-eastern Germany along the Alpine bow during these three cases. A second area of heavy precipitation is expected to stretch from the south-eastern Alps into the Dinarides mountains. From these similarities, it can be assumed that also the synoptic

fields of precipitation look similar for all three cases. Similarity is also presumed regarding the location and intensity of the rain-associated cyclones and cyclone tracks, as they strongly determine whether heavy precipitation is advected to the expected regions along the north-eastern Swiss Alps.

For 2005, both ERA-Interim and CERA-20C produce a center of heavy precipitation (up to 50 mm /24 h) in the expected region (Figure 3a and b), and tongues of heavy precipitation reach east and southeast along the Alpine

bow and along the Adriatic coast. In comparison, 20CR shows only one coherent, but larger center of precipitation that is shifted towards the south-east and has lower intensities, while representative for larger grid boxes (up to 35 mm/24 h;   Figure 3c). Variability in precipitation totals among the 56 members of 20CR is larger (interquartile range of approx. 10 to 15 mm/24 h) than among the ten members of CERA-20C (interquartile range of up to approx. 10 mm/24 h; Figure 3d and e). The cyclone fields of the 2005 case, as well as the associated cyclone tracks south

of the Alps, are well defined in the 20CR ensemble: shifts on only a couple of grid cells occur (Figure 4a, d and g). For instance, 46 members show a cyclone track at 10° E, 44° N for 21 August 2005 18 UTC. Differences to the cyclone fields in ERA-Interim are also mostly within the range of the 20CR members.

For 1910, the centers and tongues of heavy precipitation have a similar location to 2005 in CERA-20C; although intensities are lower (up to 30 mm/24 h). 20CR also shows similar centers of heavy precipitation, while intensities

in 20CR are clearly lower (Figure 3f and g). In contrast, the variability among the members is higher in the CERA-20C dataset (up to approx. 10 – 15 mm/24 h compared to below 10 mm/24 h in 20CR; Figure 3i and j). Compared to the 2005 case, the range of calculated cyclone tracks and cyclone fields for 1910 becomes larger in 20CR, and encompasses three or sometimes even more grid points (Figure 4b, e and h). A number of 56 cyclone tracks are found at two grid points (10° E, 44° N and 12° E 44° N) for 13 June 1910 18 UTC.

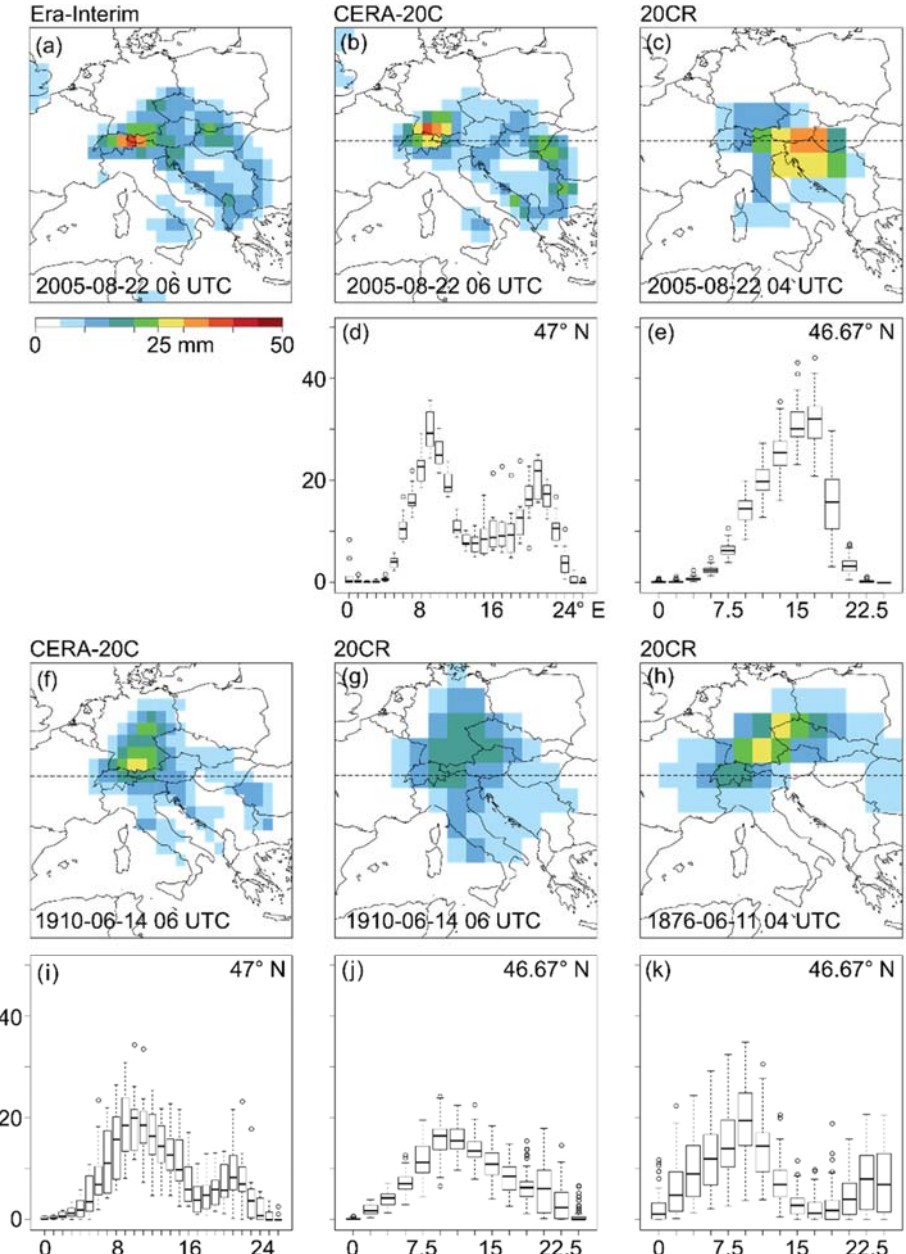

**Figure 3: Daily precipitation totals (mm/24 h; color shade) as calculated for 22 August 2005 (a, b, and c), 14 June 1910 (f and g) and 11 June 1876 (h) from ERA-Interim (a), CERA-20CR (b and f) and 20CR (c, g, and h). Slightly differing time steps are due to differing temporal resolutions of the reanalyses products. Boxplots below the map panels show,**
5     **respectively and where applicable, the variability of daily precipitation totals of all ensemble members in grid boxes along cross sections at 47° N (CERA-20C; d and e) or 47.67° N (20CR; e, j, and k). The horizontal lines in (b, c, f, g, and h) delineate these cross sections.**

For 1876, only 20CR is available (Figure 3h and k) to assess the representation of synoptic precipitation fields.
10   Compared to 1910 and 2005, the center of heavy precipitation is located more to the north-east of Switzerland, over south-eastern Germany. Intensities are higher than for 1910 and lower than for 2005. Whereas the cyclones pass across the Ligurian Sea and Northern Italy in the 2005 and 1910 cases, the bulk of the ensemble takes a more northerly path in the 1876 case (Figure 4c, f and i). A number of 25 members show a cyclone track at two grid

points just south of Switzerland (10° E, 46° N and 12° E 46° N). And while a small part of the members tracks towards the north(-east) on 12 and 13 June 1876, the rest shows a south-eastward propagation along the Adriatic Sea.

Overall, the analyses at synoptic scales (Figures 3 and 4) show that differences among the 20CR members are
substantially smaller over the region of interest (Southern and Central Europe) than over other regions of the North Atlantic / European sector (Figure 4 d, e and f); this corresponds to the relatively high density of assimilated stations over Central Europe (not shown; see Compo et al. 2015). The main fields of precipitation are approximately co-located in all three datasets (Figure 3). Variability in the 20CR ensemble is comparable to CERA-20C for the 2005 and 1910 cases. 20CR shows overall lumpier spatial patterns of heavy precipitation and lower values due to the
coarser horizontal grid, and a potential displacement of the precipitation field for the 1876 case. As expected, the uncertainty, in terms of disagreement between the 20CR ensemble members, becomes increasingly larger when going back in time. For instance, the cyclone fields and cyclone tracks over the Alpine area are only a little less well defined for 1910 compared to 2005, but much less for 1876. Among others, this is shown by the number of co-located cyclone tracks (in terms of pressure minima) in the 20CR ensemble. The algorithm detects 56 co-located
cyclone tracks at a grid point over northern Italy on 21 August 2005 18 UTC (Figure 4 g). For the time step  13 June 1910 18 UTC (Figure 4 h), a number of 56 cyclone tracks are detected at two adjacent grid points, whereas only 17 co-locations are found for 10 June 1876 18 UTC (Figure 4 i). From these analyses, we infer a very good to satisfactory positioning of cyclone tracks and cyclone fields in 20CR for the 2005 and 1910 cases, but not necessarily for 1876. This means that the boundary and initial conditions appear to be captured in 20CR for the
2005 and 1910 cases, while 1876 shows two or even more potential developments of the cyclone.

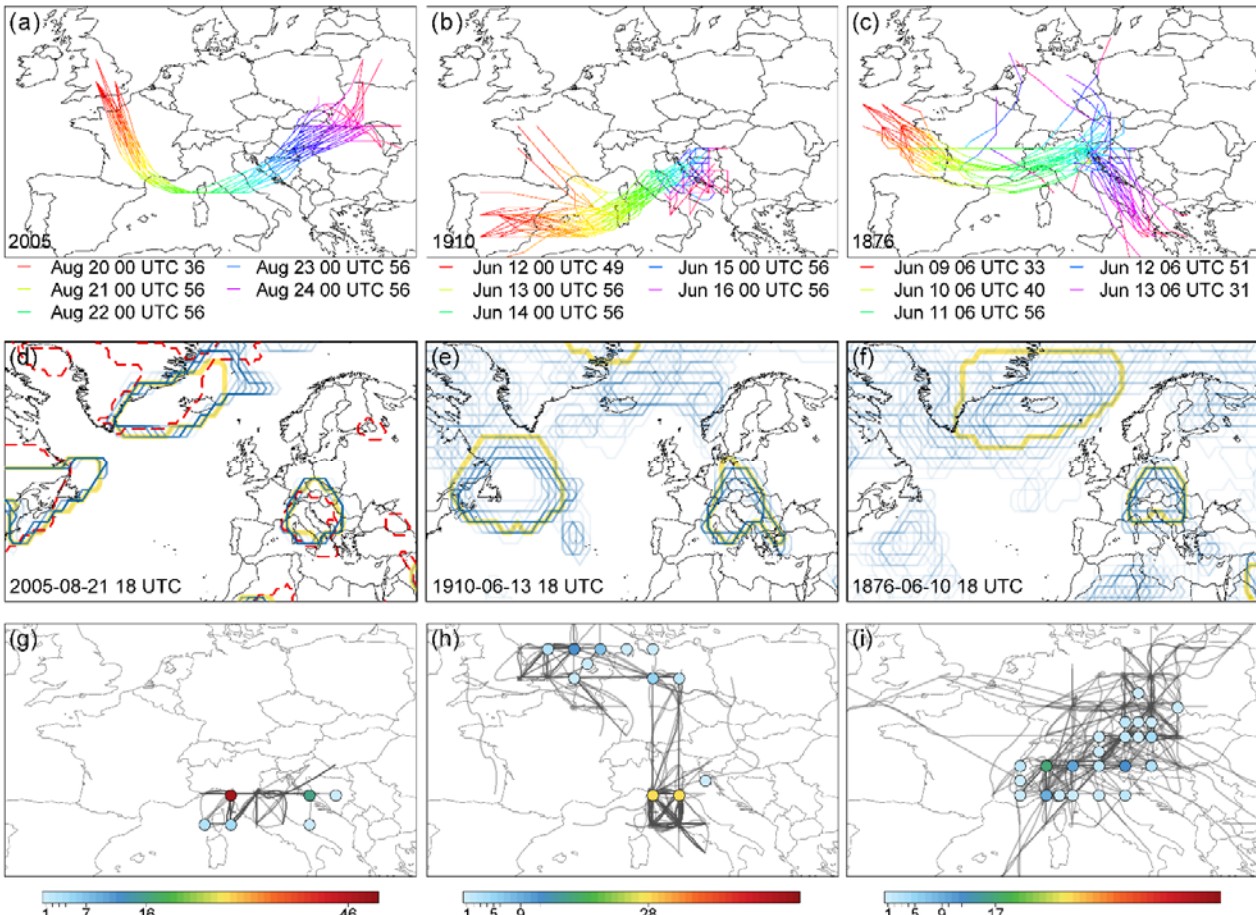

**Figure 4: Synoptic situations as depicted in the 20CR ensemble for the 2005, 1910 and 1876 cases. (a, b, c) Cyclone tracks for each ensemble member of 20CR at a mid-tropospheric, i.e. the 500-hPa level. The color of the lines corresponds to the time steps indicated below the panels. In addition, the number indicates for how many of the 56 members a cyclone position could be reconstructed at a certain time step. (d, e, f) Objectively identified cyclone fields (calculated for sea level pressure; using Wernli and Schwierz, 2006; see also Welker and Martius, 2015) in the 20CR ensemble (blue contours) and 20CR mean (yellow contours) at the time steps of (d) 21 August 2005 18 UTC, (e) 13 June 1910 18 UTC, and (f) 10 June 1876 18 UTC. The color scheme, ranging from light blue to dark blue, indicates in how many of the 56 ensemble members a cyclone is detected in the respective grid cell. The red broken lines in the 2005 panel indicate the cyclone field as calculated from ERA-Interim. (g,h,i) Cyclone centers identified for sea level pressure at (g) 21 August 2005 18 UTC, (h) 13 June 1910 18 UTC, and (i) 10 June 1876 18 UTC, the same time steps as in (d, e, f). Colored dots and the tick marks in the color key indicate the number of cyclone tracks located at a specific grid point at the respective time steps. Greyscaled lines mark the cyclone tracks over the period of 48 h before to 48 h past the respective time steps. Darker (lighter) grey shades indicate more (less) cyclone tracks along a certain path.**

## 4.2 Precipitation, cyclone tracks and moisture transport along the downscaling steps

In this section, we examine how well the three Vb cases are represented when downscaling the global information from 20CR to a 3-km horizontal grid using the WRF regional model.

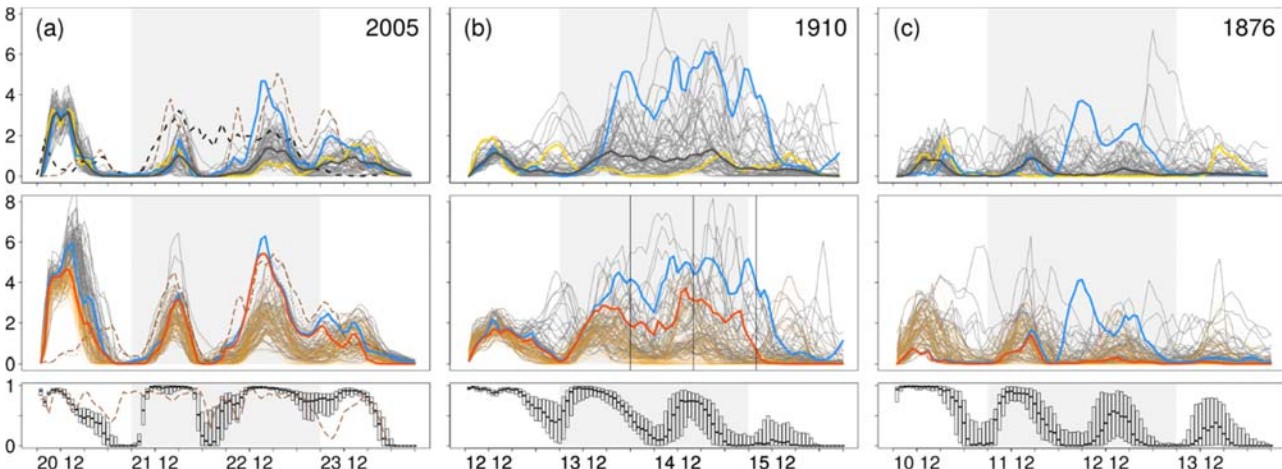

**Figure 5: Time series of simulated precipitation over northern Switzerland for (a) 20 August 2005 06 UTC to 24 August 2005 06 UTC, (b) 12 June 1910 06 UTC to 16 June 1910 06 UTC, and (c) 10 June 1876 06 UTC to 14 June 1876 06 UTC. Dark grey lines indicate mean hourly precipitation (mm/h) within the control area over northern Switzerland for the**

**56 downscaled ensemble members, as simulated for the 3-km domain (upper panels) and the 9-km domain (middle panels) . In the upper panels, the thick black (yellow, blue) solid lines mark the median (minimum-precipitation, high-precipitation) members, as selected for Figs. 6 to 9 and 11. In the middle panels, thin orange lines show the contribution of the convection parametrization to the total precipitation in the 9-km domain. The thick dark orange lines indicate convective precipitation for the selected high-precipitation members. The bottom panels show the proportion of**

**convective precipitation with respect to total precipitation in the 9-km domain. In (a), the dashed black line shows the corresponding hourly precipitation calculated from CombiPrecip, and the dashed brown lines show the precipitation from the ERAI-WRF experiment with 4 domains; the convective contribution is shown with points in the middle panel. Black vertical lines in (b) indicate the instances in time selected for Figure 11. Grey shadings mark the most intense 48-h periods of precipitation according to Figure A1.**

In a first analysis, we address the temporal variability of the simulated precipitation. Figure 5 shows time series of aggregated precipitation in the control area over northern Switzerland. In the 2005 case (Figure 5a), two distinct peaks occur on 21 and 22 August 2005 around 18 UTC. This evolution is very much in line with Hohenegger et al. (2008; their figure 8); even the increase during the second peak episode is very similar. As already seen with the standard downscaling configuration (Figure 2), intensities are mostly underestimated when compared to

CombiPrecip, and arguably, too little precipitation is produced between the simulated peak episodes. Two high-precipitation episodes are also simulated for 1910 and 1876 (Figure 5b and c), although variability among the members increases regarding the timing and intensities of precipitation. For instance, the ensemble interquartile range is smallest for 2005 (around 1.5 mm/h in the peak episode; note that this is smaller than in Hohenegger et al., 2008) and becomes larger for the earlier cases (around 2 mm/h on 14 June 1910 18 UTC, and around 2.5 mm/h on

11 June 1876 18 UTC).

For all cases, the ensemble shows most precipitation peaks in the afternoon. This would be in agreement with an enhancing effect by (embedded) convection. To investigate this effect, we turn to the second finest domain with a 9-km horizontal grid. Whereas convection is explicitly simulated for the 3-km domain, it is parameterized in the 9-km domain, resulting in WRF model variables of non-convective (RAINNC) versus convective (RAINC)

contributions to the total precipitation (the shallow convection variable RAINSH is turned off). For the 2005 case and in the 9-km domain, convective precipitation is the largely dominant process during the afternoons (Figure 5a),

reaching nearly 100 percent in all the members and on all days of the event. The proportion of convective precipitation is smaller during other times of the day and varies more in the ensemble. The same pattern is found for the 1910 and 1876 cases, although the proportion of convective precipitation is mostly smaller and variability in the ensemble is higher (Figures 5b and c).

5   Peaks of precipitation are also simulated during the initialization period of each case, which is in line with observations (not shown; cf. Stucki et al., 2012). However, the peak on 20 August 2005 is too prominent compared to CombiPrecip (Figure 5a). The convection-driven peak is simulated for all members including the minimum-precipitation member, while precipitation intensity appears more realistic in the WRF-ERAI simulation. The reason for this case-specific behavior is unknown.

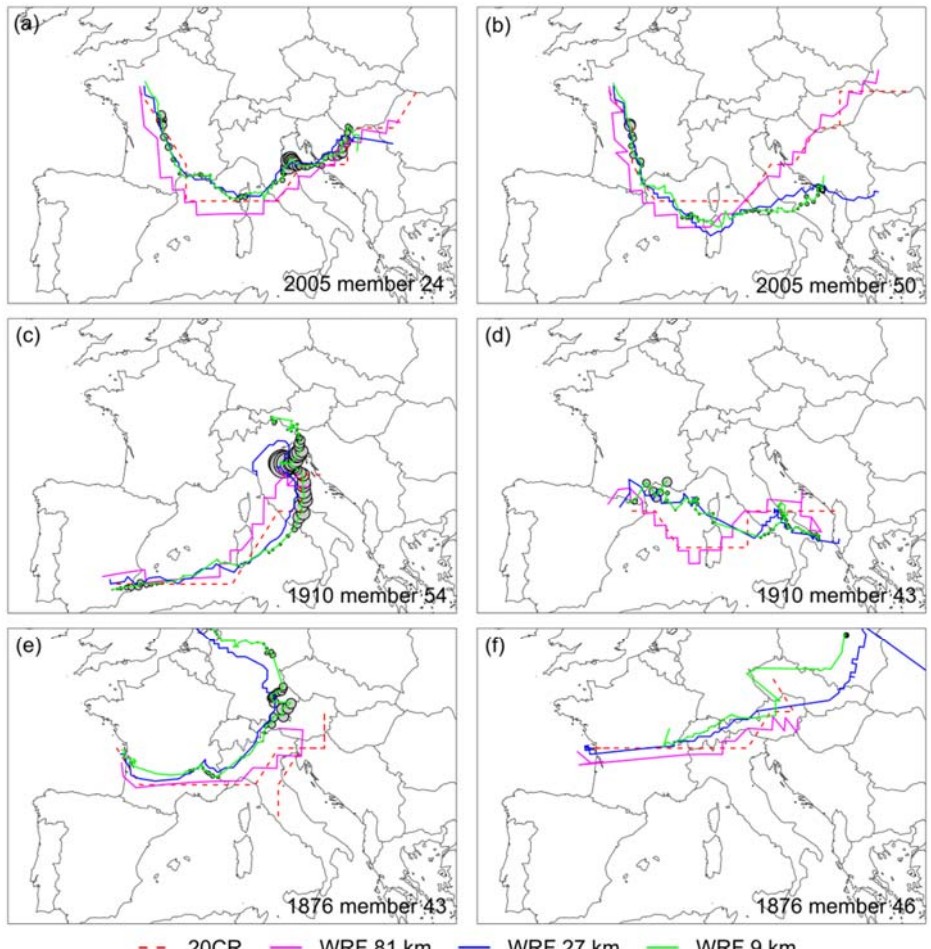

Figure 6: Cyclone tracks, calculated for 500 hPa levels, for the cases of 2005 (top), 1910 (middle) and 1876 (bottom) and for members producing maximum (left) and minimum (right) precipitation totals over northern Switzerland. The cyclone tracks in 20CR and WRF at 81-, 27- and 9-km grid sizes are shown in red, magenta, blue and green, respectively. Filled circles indicate the mean hourly precipitation over northern Switzerland at the time step when the cyclone was centered at the respective grid point in the WRF 3-km domain. The circle diameters grow linearly with the precipitation amounts; the largest circle (in c) represents 6.6 mm/h.

Next, we examine whether the found features and variabilities in the ensemble reflect differences that are already

20   present in the 20CR members (Figure 3) or if they appear along the downscaling steps. Concretely, we search for

flow features that help to systematically distinguish members with low or high precipitation simulated in the correct region (i.e. in the control area over north-eastern Switzerland). For this, we compare the simulated 48-hour precipitation totals with RrecabsD. We use the ratio of precipitation in the simulation versus the reconstructions, and the EMD between simulation and reconstruction to assess the similarity of the spatial distribution of precipitation.

For illustration, the panels in Figure A2 in the Appendix show (i) a maximum member in terms of simulated precipitation (a near-maximum member is chosen for the 1876 case because the maximum member does not show plausible patterns, cf. Fig. 5c), and (ii) a minimum member in terms of lowest precipitation totals in the control area. Indeed, the two contrasting members are exemplary for the large variability of the simulation results. Throughout the ensemble, we find members that largely misestimate the precipitation totals (the range is around 20 to 160 percent for the 2005 and 1910 cases, and 5 to 70 percent for the 1876 case, not shown), while others produce precipitation at the wrong place, but also a number of members that produce quite accurate spatial patterns and precipitation totals compared to observations and the RrecabsD reconstruction.

Figure 6 delineates the corresponding evolution of the cyclone tracks in selected ensemble members that yield maximum (a) or minimum (b) precipitation for the 2005 case. The cyclone track for the maximum-precipitation member follows closely the original cyclone track in 20CR in each downscaling step. During the peak episode, the cyclone center is located just above the Adriatic coast of northern Italy. Moreover, the multiple circles at the same location (Figure 6a) indicate quasi-stationarity of the cyclone. In contrast, the minimum-precipitation member has a cyclone track in the 27-km domain that clearly departs from the original 20CR cyclone track: Instead of recurving to the north over Italy, it keeps propagating eastward. The high-resolution domains (9-km; and 3-km, not shown) then represent refinements of these patterns without significant changes. In the 1910 case, the cyclone tracks for the maximum-precipitation member (Figure 6c) also show the vicinity to 20CR in all downscaling steps, the recurving to the north and the same location during the stalling, i.e. peak episode. In contrast, the minimum-precipitation member shows a more southerly and eastward track after it reaches Italy (Figure 6d). That is, the tracks of 20CR and the coarsest downscaling step never turn towards the north, thus, making it more difficult to bring precipitation towards the target area. In contrast to the 2005 and 1910 cases, the algorithm has difficulties to detect clear cyclone tracks along the downscaling steps for the 1876 case (Figure 6e and f). In addition, the found tracks run just south of or even across Switzerland, hence on a much more northerly path than for the other two cases. Such tracks do no longer represent a typical Vb trajectory.

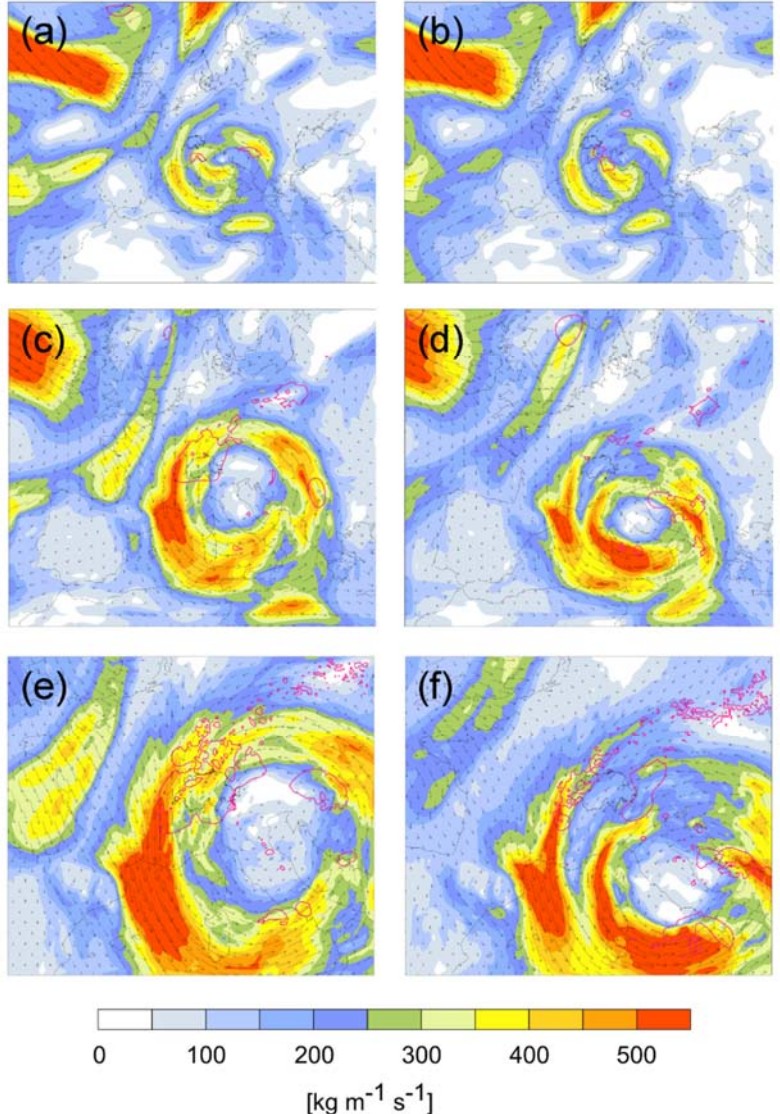

**Figure 7: IVT (kg m$^{-1}$ s$^{-1}$) on 22 August 2005 18 UTC for in the WRF 81-km (a, b), 27-km (c, d), and 9-km (e, f) domains and for a member producing maximum (#24; a, c, e) and minimum (#50; b, d, f) precipitation totals. Smoothed hourly precipitation is shown with pink contours of 0.2 and 1mm/ h.**

The panels in Figure 7 (for the 2005 case), Figure 8 (for 1910), and Figure 9 (for 1876) illustrate the link between the exact position of the cyclone track and the moisture transport in terms of IVT, showing variations of synoptic to meso-scale features along the downscaling steps for the simulated peak times. In the 2005 case, the IVT patterns of the vortex located south-east of Switzerland are very similar among the two contrasting members in 20CR (not shown), and small differences appear in the 81-km domain (Figure 7a and b). This changes in the 27- and 9-km

domain (Figure 7c and d): In line with the cyclone track in Figure 6, the IVT vortex of the minimum-precipitation member is clearly shifted towards the south. The location of the cyclone center strongly determines the intensity of the moisture flux and precipitation over the Alps: In the maximum-precipitation member in Figure 7e, moisture is transported all the way around the Alpine chain and mainly over its northern side. In contrast, for the minimum-precipitation member in Figure 7f, the circle of intense IVT around the cyclone center is shifted southerly and is

moreover partly interrupted over the northern Alps / Switzerland, arguably because a lot of the moisture already precipitates upstream, i.e. over the Dinarides mountains in the Balkans.

In the 1910 case (Figure 8), the corresponding patterns of the IVT vortices are very similar to the 2005 case, although showing lower intensities over the Alps in 20CR (not shown) and the 81-km domain: The maximum-precipitation member shows northerly winds and intense moisture flux / precipitation over Switzerland, associated with the cyclonic IVT pattern surrounding the cyclone center (Figure 8c and e). In the minimum-precipitation

5      member however, the cyclone center and associated cyclonic IVT pattern are shifted south-eastwards, such that the intense IVT misses the Central Alps (Figure 8d and f). In the 1876 case (Figure 9), the maximum-precipitation member features hardly any structures of a vortex in 20CR (not shown) and in the 81-km WRF domain, and the IVT vortex appears broken and misplaced at higher resolutions. A center of the cyclone is located over northern Germany and induces intense moisture transport in a westerly flow. Hence, some areas in northern Switzerland

10     receive intense precipitation, although it is not anymore associated with a classical Vb cyclone. In the minimum-precipitation member, the cyclone center is also located to the north, but also to the west of Switzerland (Figure 9d and f). Again, Switzerland is on the south side of the vortex, within southwesterly moisture flux reaching into western Switzerland only, and missing most of the Central Alps.

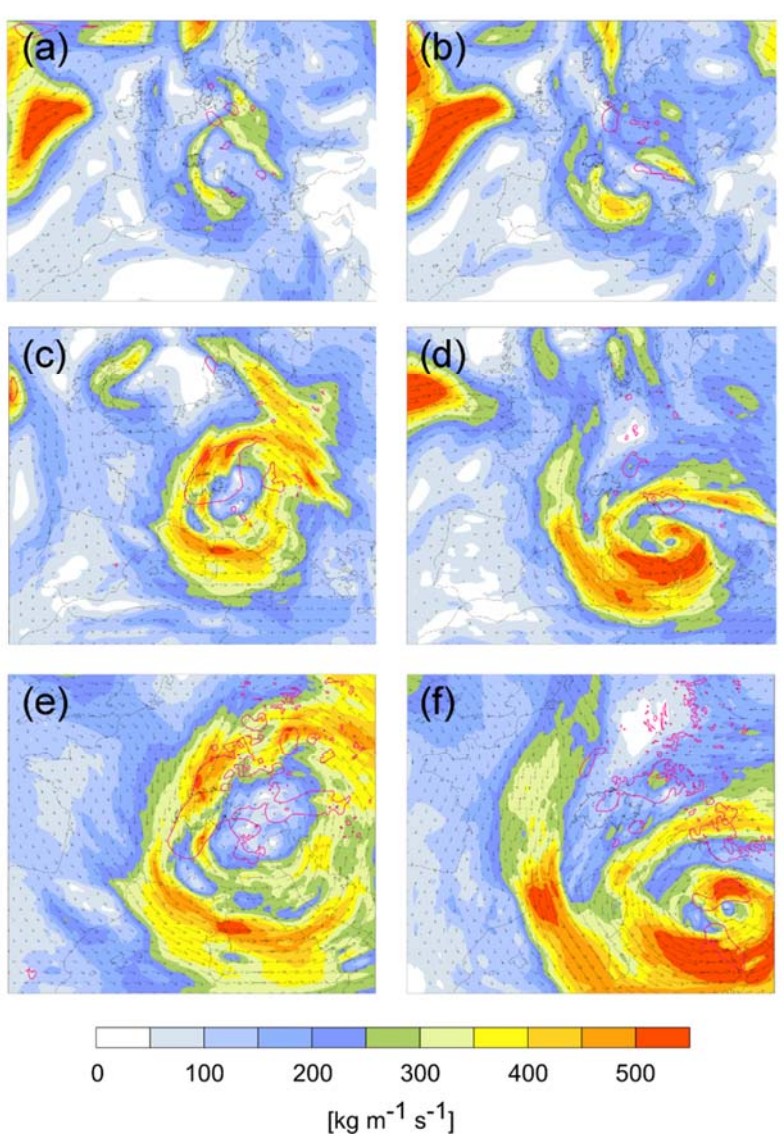

**Figure 8: As in Figure 7, but for 14 June 1910 18 UTC, and for different maximum (#54) and minimum (#43) members.**

Furthermore, Figure 10 demonstrates that indeed, precipitation intensity over northern Switzerland was closely related to the intensity and direction of the moisture transport towards the Alps during the heavy-precipitation phase of the three cases. In the 2005 and 1910 cases (Figure 10a and b), IVT intensities of more than 200 kg m$^{-1}$ s$^{-1}$ are

5    advected from Northwest to Northeast (i.e. directed towards the northern side of the Alps). This is concurrent with average precipitation rates of up to 8 mm/h, whereas precipitation rates become clearly lower with decreasing IVT and with other inflow angles, as seen in the 1876 case (Figure 10c). Similar results were found by Froidevaux and Martius (2016).

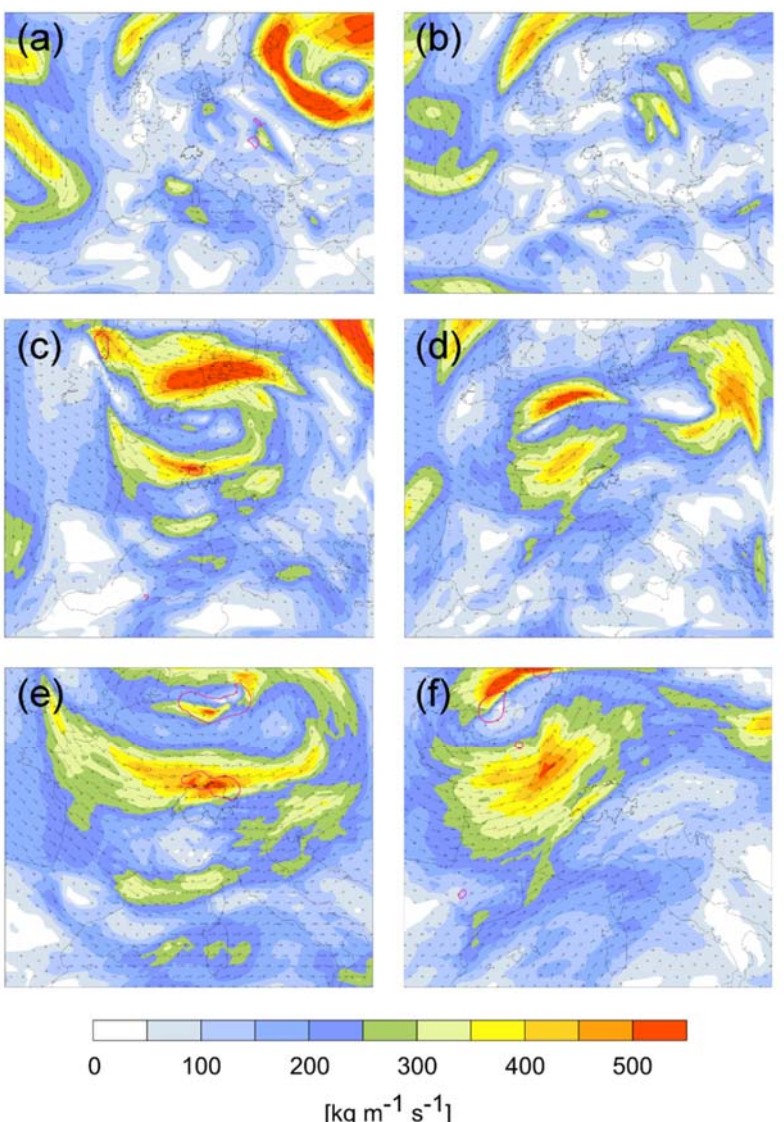

**Figure 9: As in Figure 7, but for 13 June 1876 00 UTC, and for different maximum (#43) and minimum (#46) members.**

In summary, the 2005 and 1910 cases behave similarly along the downscaling steps in the simulation, whereas the 1876 case deviates in a range of aspects. In the 2005 and 1910 cases, Vb cyclones exist for all members in 20CR

15    (Figure 4, Figure 6), and the cyclone centers cross the WRF domains at 81-, 27- and 9-km grid sizes. In the 3-km domain, the trajectory of the cyclone centers typically passes southwards of the domain, and a clear cyclonic

circulation is systematically present, corresponding to the position of the cyclone in the 9-km domain (Figures 7, 8 and 9). This can be expected, as the cyclone is larger than the two smallest domains, which hampers shifting of cyclone centers. Moreover, we find that the cyclones with centers that stall over a specific location of Northern Italy / the Adriatic Sea are associated with more intense precipitation over north-eastern Switzerland. The maximum-precipitation simulation for 1910 produces even larger totals than observed (Figure 5, see also Figures 8, 10 and A2 in the Appendix). This may indicate that under slightly different atmospheric conditions, e.g. with longer stalling of the cyclone at a particularly unlucky location, the real cases could have had even worse impacts.

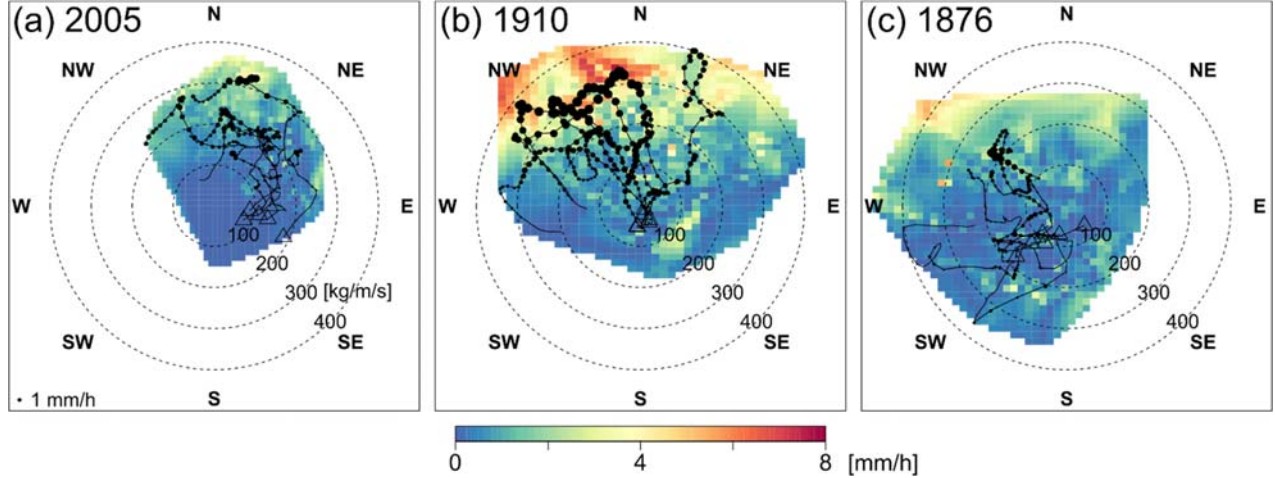

**Figure 10: Hourly precipitation (mm/h; color shade) as a function of IVT (kg\*m-1\*s-1; dashed circles) over north-eastern Switzerland during 48-h periods starting at (a) 21 August 2005 06 UTC, (b) 13 June 1910 06 UTC, and (c) 11 June 1876 06 UTC. Both precipitation and IVT are averages over the control area in north-eastern Switzerland in the 3-km WRF domain. The 480 black dots represent 48 different time steps for a subset of ten members (see Section 3.1). The size of the dots represents precipitation intensity; the location of the dots on the radial diagram represents IVT: the azimuth represents the direction of IVT and the radial component its intensity. All time steps of a same member are connected by a line, with the last time step marked by a triangle. The corresponding two-dimensional interpolation of the precipitation intensity is shown in color. The color maps hence represent the mean precipitation intensity as a function of IVT intensity and direction.**

In contrast, members with more southerly tracks do not produce heavy precipitation in this region. With a displacement to the south, the moisture transport does no longer provide a northern inflow towards the Alps, which then inhibits the orographic lifting along the Alps (Figures 7, 8 and 9; and Figure A2 in the Appendix). Hence, the moisture removal from the atmosphere is limited, leading to less precipitation in general and especially in the target area.

Too northerly tracks are not helpful in generating plausible precipitation patterns either. The 1876 case shows that the vortex structure is destroyed as soon as the cyclone centers are located too close to the Alps. Instead of intense moisture advection from a sector North, advection on the north side of the Alps shifts to a sector West or even South. This behaviour of the model can be explained by the interaction of the Alpine orography with atmospheric circulation. Indeed, Vb cyclone trajectories are typically initiated by deepening upper-level troughs, which finally cut off from the westerly flow when passing over the Alps (e.g. Awan and Formayer, 2017). The interaction of upper-level troughs with the Alpine orography have been described in detail (Buzzi and Tibaldi 1978; Aebischer

and Schär 1998; Kljun et al. 2001: the underlying processes include flow splitting and lee cyclogenesis, with further amplifications of the cyclone formation by frontal retardation and latent heat release due to orographic lifting. The combination of these processes implies that the cyclones are formed on the lee of the right side of the Alps, typically over the Ligurian Sea. In 20CR, however, the Alpine orography is very coarse, smoothed and reaches only about

1000 m a.s.l. (cf. Stucki et al., 2012). Hence, the influence of the Alps on the large-scale flow is limited in 20CR. Given also that the 1876 case is least confined by pressure observations, this allows untypical cyclone tracks in many 20CR members. Once accounting for a more and more realistic orography throughout the downscaling steps with WRF, the high-resolution runs may thus end up in a compromise simulation - driven both by the WRF model physics and by the 20CR input flow. In other terms, the large-scale flow forced from 20CR might not be compatible

with the orography of the high-resolution domains.

To conclude the analyses, Figure 11 illustrates how a certain combination of cyclone tracks and cyclonic moisture flux translates into a specific weather situation at the surface. For this, we select the maximum-precipitation member (#54) for the 1910 case and show an early, mid-, and late instance of the heavy-precipitation period (cf. Figure 5), and we compare it with findings regarding the 2005 case.

For 14 June 1910 00 UTC, the 3-km downscaling shows patches of heavy precipitation along the Alps and Alpine foothills of northern Switzerland (Figure 11a). Many of them appear in banded structures, similar to findings from the 2005 case (Bezzola and Hegg, 2007; Langhans et al., 2011). The structures are generally oriented parallel to the Alpine bow and reach from south-eastern Germany into central Switzerland. Surface winds in the control area come from sector North to North-Northwest and weaken upwind of the Alpine barrier. Concurrent areas of

convection appear in the 9-km simulation (Figure 11b), and the pressure gradient along the Alpine rim shows the Vb low-pressure system over the Adriatic Sea. At the same time, the IVT vortex just starts to show intense moisture transport from the northeast towards Switzerland (not shown, cf. Figure 7 for a later instance). In all, this indicates persistent airflow upon orography and orographic lifting, as documented for the 2005 case (Zbinden, 2005; Bezzola and Hegg, 2007) and visible in CombiPrecip and wind observations for the evening of 21 August 2005 (not shown).

On 14 June 1910 16 UTC, the center of the low-pressure system is located just south of Switzerland (see red dot in Figure 11d). Accordingly, the northerly cross-Alpine flow is substantially stronger, and heavy precipitation becomes most intense along the northern Alpine ranges (Figure 11c). Again, this shift of the heavy precipitation into the Alpine ranges with enhanced northerly flow is in line with analyses of the 2005 case (Zbinden, 2005; MeteoSwiss, 2006; Bezzola and Hegg, 2007), and with CombiPrecip and wind observations for 22 August 2005

(not shown). The 9-km simulation shows the associated areas with intense convection reaching into Switzerland from the northeast (Figure 11d). At this stage, the cyclonic flow forms a distinct arc that stretches from the eastern to the central Alps.

On 15 June 1910 08 UTC, the SLP minimum has crossed the Alps to the northeast of Switzerland (Figure 11f). In the 3-km simulation, heavy precipitation occurs in an area of southwesterly winds along the Alps, the Swiss Plateau,

and towards the east, while the flow remains north-westerly at higher elevations and towards the (north)west (Figure 11e). Concurrently, the SLP fields in the 9-km simulation indicate higher pressure from the west, before precipitation intensifies along the eastern Alps, while it finally eases over Switzerland. This is again analog to a late stage of the 2005 case (MeteoSwiss, 2006; Bezzola and Hegg, 2007; Figure 11f), e.g. visible in CombiPrecip and wind observations for around midnight on 22 August 2005 (not shown).

In the end, all simulated instances are associated with heavy precipitation over northern Switzerland, with slight changes in the inducing weather dynamics. In the first instance, banded convection and orographic lifting both contribute to intense precipitation. The second instance, corresponding to the peak precipitation, is associated to stronger northerly winds and a distinct cyclonic moisture flux around and over the Alps. The last instance is linked to a shift towards westerly advection and increasing pressure. While the early stage of the SLP cyclone track calculated for member 54 is not typical (no cyclogenesis in the classic Genoa region, see Figure 6c), the surface analyses of the two innermost domains during the heavy precipitation phases show that the simulation produces realistic near-surface weather dynamics at local scales, and they can clearly be related to the circulation and features of a Vb cyclone.

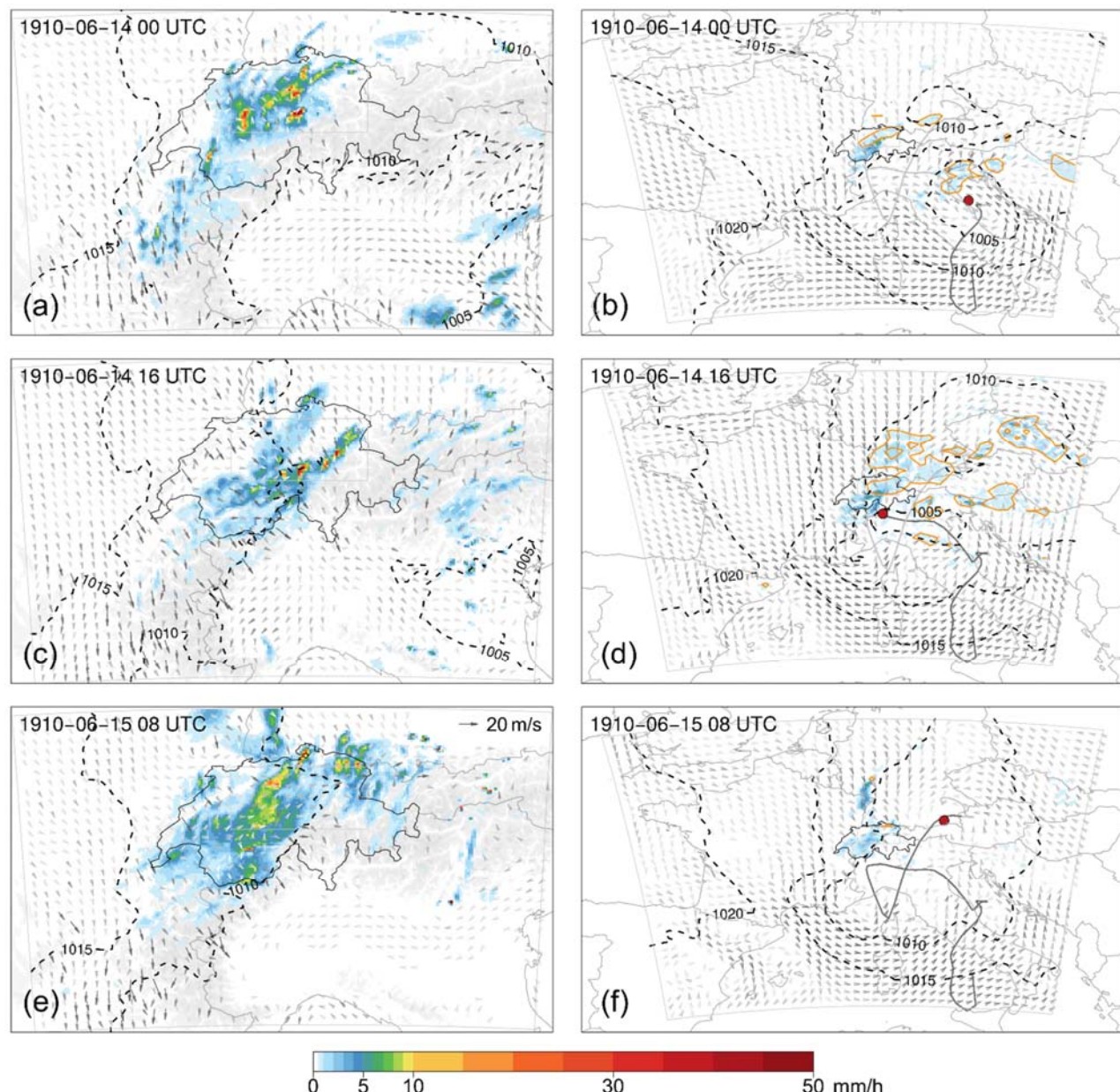

**Figure 11: The 1910 event (as shown in the downscaled member #54) in the 3km WRF domain (left) and corresponding 9km WRF domain (right) and for three instances in time corresponding to heavy precipitation over northern Switzerland, that is at (a, b) 14 June 1910 00 UTC (c, d) 14 June 1910 16 UTC, and (e, f) 15 June 1910 08 UTC. Hourly precipitation (mm/h; color shade), 10-m wind field (m/s, light grey vectors darken with increasing velocity, see**

**reference vector of 20 m/s in panel e) and SLP (dashed contours, in 5-hPa increments). In (b, d, and f), orange contours show the contribution of the convection parametrization to the total precipitation (contours at 2 and 5 mm/h). The dark (light) grey lines in the 9-km domain indicate the smoothed cyclone track over starting on 13 June 1910 00 UTC and until the shown instance. The red dot marks the cyclone center at the respective instance in time.**

## 5 5 Summary and conclusions

In this study, we have assessed the potential of dynamical downscaling from 20CR input to 3-km grid sizes for three well-known Vb cyclones that led to heavy precipitation and flooding in (north-eastern) Switzerland in August 2005, June 1910 and June 1876. In particular, we have analyzed the sensitivity of the produced precipitation totals in a control area in north-eastern Switzerland to (i) the setup of the regional weather model and to (ii) the

representation of moisture flux in the 20CR ensemble and along the downscaling steps.

Regarding the configuration of the regional weather model (WRF in our case), we find that the downscaling procedure with a standard setup results in mostly underestimated precipitation totals, and a large variability among the ensemble members. This has led to a series of experiments to test the sensitivity of precipitation totals to a range of differing model setups. For our purposes, we have found that short spin-up periods (encompassing around 24

hours before the heavy-precipitation episode) are preferable over long spin-up periods, which would allow (partial) adaptation of small-scale and slow-reacting variables in the model, such as soil moisture, for instance. In our experiments, precipitation totals in the ensemble become less variable and more realistic if the cyclones are already present in the outermost model domain at model initialization; if not, the simulation runs too freely. Other than that, substantial changes of standard physics options do not increase model performance, be it cumulus or micro-

physics, or two-way nesting. Given that the parameterizations are turned off for the 9-3-km downscaling step, comparable outcomes with differing physics schemes point to the importance of the larger-scale atmospheric flow for producing the heavy precipitation. Although we find no relevant enhancements from nudging in smaller domains in our test experiments, nudging smaller domains could still be beneficial for other specific studies. In the simulations of the cyclonic vortex, the largest deviations along the downscaling steps occur in the 27-km domains.

The increasing variability of the simulations in these domains might be explained by the fact that no nudging is applied, while it is in the larger, 81-km domains. Although going back far in time, we have only analyzed a very small number of events – many more cases would be needed to reach robust recommendations on how to configure a model for Vb cases. Nevertheless, we have demonstrated that that one can achieve a relatively best configuration for the desired application with a well-thought series of experiments.

In our context, the EMD has proven to be a valuable and intuitively understandable tool for spatial verification of the simulated precipitation fields with observation-based reconstructions. In fact, our EMD analysis results in similar rankings as obtained from the more common spatial verification scores and metrics (MAE, box ratios; see Table 2) or from intersubjective, visual analyses of the precipitation patterns.

Regarding the representation of precipitation and related variables in the 20CR ensemble, we find that 20CR

delivers a well-confined ensemble for the 2005 case. Given the coarser horizontal grid sizes and lower vertical resolution, it compares well with other long-term reanalysis products. The 1910 case is also comparably well defined in the 20CR ensemble, whereas the 1876 case shows more uncertain developments of the cyclone fields in the ensemble. This gradually increasing uncertainty when going back in time is also found for precipitation-related

variables along the downscaling steps. For instance, the dynamical downscaling procedure captures the peak episodes of all three cases, although gradually less well going back in time. Furthermore, the accuracy of precipitation totals is closely linked to the exact cyclone track and the exact location of the vortex when it comes to stalling. Concretely, this location should be over northern Italy, or just off the northern Adriatic coast for best simulation results with regards to the intensity and spatial distribution of precipitation totals over north-eastern Switzerland.

Ensemble members that do not follow such a trajectory produce erroneous precipitation totals in the control area, where too southerly tracks generally produce too little precipitation, and too northerly tracks lead to a break-up of the associated vortices because of interaction with the Alpine (model) topography. This is found to be a decisive element, because the exact (stalling) location of the vortex strongly influences the cyclonic moisture transport around the Alps and the exact inflow angle from a sector North to the Central Alps. In fact, IVT intensities of >200 kg m-1 s-1 or even more from the right direction are needed to reproduce the extreme events. Interestingly, we have found a range of members that produce more precipitation than observed and reconstructed for the 1910 case. We infer that with a slightly different, hence ideal constellation of the cyclonic vortex to produce heavy precipitation over northern Switzerland, e.g., a longer stalling at the right location, the 1910 floods could have had even worse impacts.

Misplacements of the vortex increase in the ensemble from the 2005 to the 1910 and 1876 cases. While the patterns and dynamics can be reproduced for the 2005 case and, a bit less well, for the 1910 case with downscaling from 20CR, the variability of the cyclone fields and tracks becomes very large in the 20CR ensemble for the 1876 case. As a consequence, we find synoptic patterns in some members that are substantially different from the 2005 case, e.g., with some cyclone tracks that do not anymore follow a typical Vb path anymore. Furthermore, the increasing uncertainties in the ensemble going back in time are also due to the decreasing quality and amount of assimilated pressure data in 20CR. For illustration, the total number of stations assimilated in 20CR in the year 1876 is 218 (Compo et al., 2015). This number grows to 377 in the year 1910 and to 9251 in the year 2005. Of course, this uncertainty propagates into our downscaled ensemble. In turn, this means that with the 1876 case, we may have reached the limits of downscaling from the current 20CR (version 2c with a 2° by 2° horizontal grid) for such complex weather situations. The WRF regional model requires more accurate locations and intensities of input variables, like cyclone fields and moisture transport, to properly reproduce such sensitive Vb cases. On the upside, we have shown that despite of these deficiencies, single ensemble members, even from the early cases, can be used to analyze and illustrate local-scale weather dynamics, as well as sensitivities of the precipitation over northern Switzerland to the evolution of the associated Vb vortex.

The question remains whether a full ensemble needs to be downscaled to gain such insights. Generally speaking, the benefit from downscaling all 20CR members is that we obtain a full set of propositions for local weather patterns during historical events. In terms of impact and intensities (in our case the local precipitation totals over northern Switzerland), the spread between these propositions is very large, reflecting the strong uncertainty inherent to the process of downscaling over a wide range of scales (here from 200 to 3 km). Using ensemble members hence has allowed us to (i) compare members with observations and select realistic runs, and to (ii) relate the differences among the members in local weather to a different evolution on larger scales. In hindsight however, the limitations of downscaling and the potential ranges of the precipitation-related variables may as well be predictable from the

input data to some extent. In our case, the well (or, in contrast, badly) confined cyclone tracks and fields in 20CR for the 2005 and 1910 (1876) cases give a good indication regarding the prospects of success for dynamical downscaling. This means that in a case where the driving atmospheric dynamics on a large scale can be anticipated, the chances of a good reproduction of the local patterns and intensities with accordingly selected ensemble members

are high (cf. Stucki et al. 2015). A second option would be to save computational costs by downscaling to an intermediate scale in the first place, assess the relevant dynamics in this domain, and then do the full downscaling with a well-reasoned selection of members. In our case, the largest deviations from the initial conditions often appear in the 27-km domain (the largest domain without nudging), if not already present in the 20CR member. This means that downscaling to the first non-nudged domain could be sufficient to assess if an ensemble detects a

cyclone well. In such a way, future studies may minimize the computational efforts for downscaling from a coarsely resolved reanalysis ensemble.

**Appendix A**

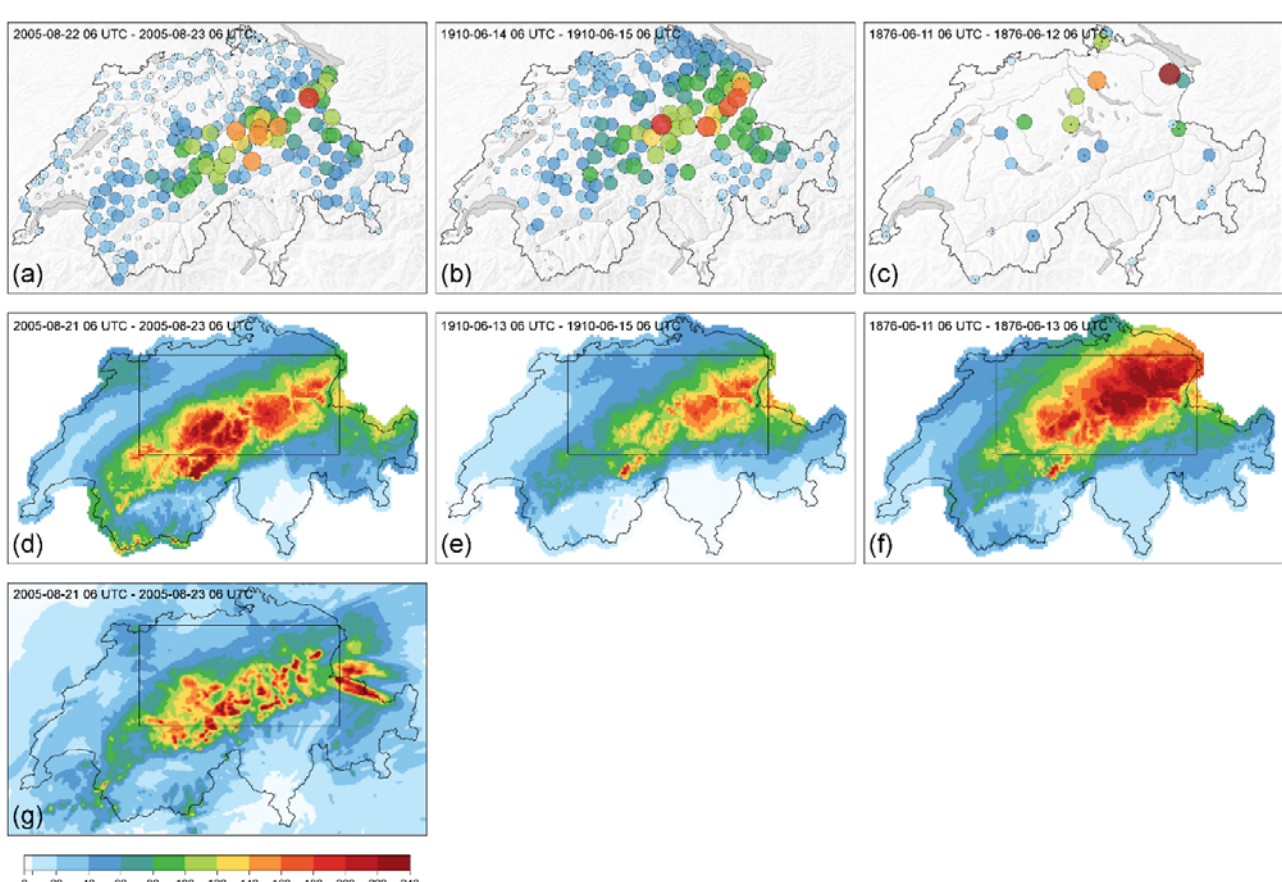

**Figure A1: Upper panels: Daily precipitation totals (mm/24 h, indicated by color shade and circle sizes) from**

**measurements over Switzerland, starting on (a) 22 August 2005 06 UTC, (b) 14 June 1910 06 UTC, and (c) 11 June 1876 06 UTC. Middle panels: Reconstructions of 48-h totals starting at (d) 21 August 2005 06 UTC, (e) 13 June 1910 06 UTC, and (f) 11 June 1876 06 UTC, as derived from RrecabsD data. Lower panel: Reconstruction of (g) 48-h totals starting at 21 August 2005 06 UTC as derived from CombiPrecip. The rectangle inset shows the smaller box (i.e., control area) used for the VarRatio, EMD and precipitation totals; the full panel shows approximately the larger box used.**

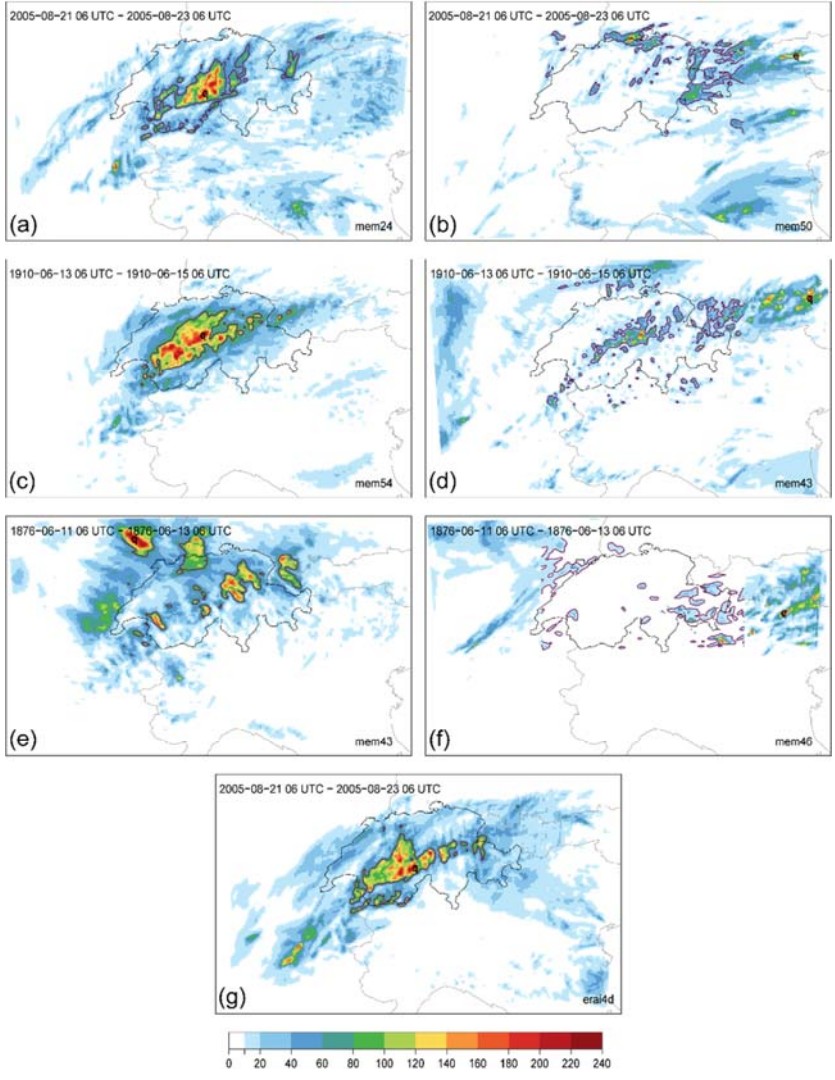

**Figure A2: 48-h precipitation totals (time periods as in Figure A1 d, e, and f) for (a,c,e) maximum-precipitation members and (b, d, f) minimum-precipitation members for (a,b) the 2005, (c,d) the 1910 and (e,f) the 1876 cases, and for (g) the same simulation setup, but with downscaling ERA-Interim for the 2005 case. Precipitation totals may be larger than 240 mm in the panels.**

### Author contribution

PS, PF, MZ, MM and AM designed the experiments. MZ and MM carried them out. FAI contributed RrecabsD. PS, PF, and MZ produced the figures and tables. All authors contributed to interpretation of the analyses, particularly the spatial verification, as well as writing or reviewing the manuscript.

### Acknowledgements

PS and PF have been supported by the Oeschger Centre for Climate Research, University of Bern. MZ has received support from the Federal Commission for Scholarships for Foreign Students through the Swiss Government

Excellence Scholarship (ESKAS No. 2015.0793) for the academic year(s) 2015-2018/19. Support for the Twentieth Century Reanalysis Project dataset is provided by the U.S. Department of Energy, Office of Science Innovative and Novel Computational Impact on Theory and Experiment (DOE INCITE) program, and Office of Biological and Environmental Research (BER), and by the National Oceanic and Atmospheric Administration Climate
Program Office.

## Competing interests

The authors declare that they have no conflict of interest.

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
