# Peer review of "Simulations of the 2005, 1910 and 1876 Vb cyclones over the Alps – Sensitivity to model physics and cyclonic moisture flux"

_Natural Hazards and Earth System Sciences, 2019_

## Referee Comment (RC1) · Anonymous Referee #1 · 27 Jun 2019

General comments:

The authors downscale three historic flood events caused by so-called Vb-cyclones and analyse the conditions under which the high-resolution simulations capture precipitation over Switzerland best. They compare model set-ups with different initialization periods, parametrizations and nudging. The most important factor determining the result with respect to precipitation is the correct representation of the cyclone path.

The article is well written and structured. It addresses three extreme precipitation events and is therefore within the scope of the journal. The authors systematically analyse different model configurations and initial data sets which allows them to identify

influencing factors of higher and lower importance. The result is a good reference for other scientists applying dynamical downscaling to study extreme precipitation events. It also shows the pros and cons of using 20CR reanalysis for initializing regional model simulations, as it turns out that deviations to observations increase for events lying further in the past.

There is however one aspect that is missing in the paper. The authors downscale to a precipitation resolving resolution of 3 km. As a motivation they cite Zängl 2007 and Prein et al. 2015 who analysed the 2005 event (page 3, line 29). The simulations performed by Stucki et al. would allow to study the effect of convection resolving simulations versus simulations with convection parametrization also for the two other cases. The paper would strongly benefit from adding an additional section which discusses the effect of the last downscaling step in more detail.

Specific comments:

p.2 l.4: The cited article (BAFU 2005) is missing in reference section.

p.2 l.12: Cyclogenesis of Vb cyclones can be outside the Mediterranean region but most of them pass the Golf-of-Genoa region (e.g. Messmer et al., 2015).

Table 2: Does a positive mean absolute error means that the model initialized with 20CR produces more precipitation than observed and that initialization with ERA40 (p.11) overestimates precipitation even further? This is probably not the case as it would be in contrast to Fig. 2. Please define what a positive MAE value indicates. From looking at Fig. 2 I also don't understand how MAE48h can be 27.75 for sp10 when averaging over 10 ensemble members. Shouldn't the error be much higher?

p.11 l.28: Please give a short explanation how cyclone fields are calculated (1-2 sentences). To understand what is shown one shouldn't need to read another paper. Please also clarify if cyclone fields and cyclone tracks are both calculated at SLP.

p.12 l.17 +others: You use storm track and cyclone track synonymously. This is con-

fusing as you only show cyclone tracks. The term storm track is mostly used for the variance at synoptic time scales (e.g. Hoskin and Hodges, 2002: New perspectives on the Northern Hemisphere winter storm track. Journal of Atmospheric Sciences, vol. 59, Issue 6,

pp.1041-1061). I suggest that you change "storm track" to "cyclone track" everywhere in the manuscript.

Fig.4 first row: Genoa cyclogenesis is not visible on these figures.

Fig.4 middle row: There is a time step indicated in each panel but the caption suggests that different time steps are combined. This is confusing.

Fig.4 last row: Please redraw. It is almost impossible to see the cyclone tracks. In addition, the caption states "as in middle panels" even though g, h and I show a sequence of days while the d, e and f show single time steps. I also don't understand why the numbers on the colour bars differ for g, h and i.

Fig. 5a: There is a very prominent precipitation peak before the event starts. The text does not give any explanation for this peak.

Figs 7, 8 and 9: Are the black dots supposed to indicate the land sea mask? This does not come out neither in print nor on my screen. Please redraw.

p.24 last paragraph: The decreasing differences to observations with time in 20CR are probably due to the increasing quality/amount of data that is assimilated in 20CR. It would be good to mention how the input data for the 20CR product has changes between 1976 and 1910.

Fig. A2 caption: "(time periods as in Figure 2)". Fig. 2 only shows 2005 Fig. A2 shows all 3 episodes.

Technical corrections:

p.9 l.11: Value in the table is 0.59 the value in the table captions is 0.49 (sp10 48-hour

precip). Which one is correct?

---

## Referee Comment (RC2) · Anonymous Referee #2 · 24 Jul 2019

Simulations of the 2005, 1910 and 1876 Vb cyclones over the Alps – Sensitivity to model physics and cyclonic moisture flux
Peter Stucki et al.
nhess-2019-174

There are 3 main objectives of this paper (i) to find a setup of the WRF model that is adequate for dynamical downscaling from 20CR, (ii) to investigate sensitivity of heavy precipitation to cyclonic moisture flux and (iii) to assess the uncertainty along the downscaling steps and among the ensemble members for historical cases.  The paper is well written but I'm concerned about the balance of the paper.  The paper is dominated by the technical aspects of performing downscaling for historical events (with poorly-defined motivation for performing model tests and testing ensemble suitability) and contains too brief analysis of the cyclonic moisture flux to achieve objective (ii) (see general comments below).  If the points below are addressed this paper would be suitable for publication in NHESS.

General comments

1.  I'm concerned that there is no motivation given for testing sensitivity to the convection, microphysics schemes or nesting in WRF.  What was the reason for performing these simulations?  Did the authors have a hypothesis that they wanted to test?  What are the differences between the schemes?  The authors conclude that there is no difference in performance when changing the cumulus scheme or nesting but do not analyse this result.  Did they expect to see a difference?  If so, why are the results insensitive to these choices?  What is the conclusion of these experiments and how general are they, i.e. would the same hold for other historical cases or are they specific to these cases?  If the conclusions are case specific, then perhaps this analysis could be reported in an appendix?

2.  There is some confusion in the paper over what constitutes a 'good' ensemble spread. The answer to this depends on the hypothesis being tested. At some points in the paper the authors claim the ensemble is good or bad by examining spread in the precipitation totals (figure 2, table 2) concluding that the 10-day forecast runs 'too freely' because the precipitation accumulation spread is large.  However, later in the paper they examine the spread in cyclone tracks (figure 4) and conclude that there is a 'good' spread in storm track position for the 2005 and 1910 cases but not the 1876 case (meaning smaller track position differences in the ensemble).  If the focus of the paper is to test the sensitivity of precipitation accumulation over Switzerland to cyclonic moisture flux, some spread in precipitation accumulation is surely necessary?  However, spread in precipitation that occurs due to factors such as cyclone position presumably need to be minimised?  Is this the rational for the later measure of ensemble suitability?  If so, why is precipitation spread used in the early analysis of ensemble spread? There appears to be some inconsistency in the analysis of ensemble suitability in the paper which needs to be clarified.

3.  Analysis relating to the sensitivity of precipitation to cyclonic moisture fluxes (figures 7-10) is described in just 19 lines.  This is rather brief for 4 figures, especially given that one of the major objectives of the paper is to examine 'sensitivity to cyclonic moisture flux' (title). The analysis must be expanded to provide a better balance

between the technical aspects and the scientific hypothesis testing analysis and to achieve objective (ii).

Specific comments

1. Page 1, line 22: What is 'moderate' spectral nudging?
2. Page 2, line 20: How is the moist air 'let' around the Eastern Alps? Do you mean advected?
3. Page 4, line 28: The authors state that they use a 'consistent part of the calibration period, which is accordingly slightly reduced'. I'm unclear what the consistent part of the calibration refers to. Please could the authors expand on this?
4. Page 5, line 5. It would be useful to know if any of the assimilated surface pressure observations were located in Switzerland.
5. Figure 1: The numbers on the colour bar have been cut off.
6. Page 6, line 26 and page 16, line 18: The authors refer to 'two peak episodes' but in figures 2 and 5 the CombiPrecip dataset does not show 2 peak episodes. Instead there is continuous high precipitation rates over a 30hr period.
7. Figure 2: The right-hand axis does not have any units. Also, it is not clear what the red numbers represent.
8. Page 7, line 1: It is not surprising that a 10-day forecast exhibits large spread in the ensemble. However, this is not necessarily a bad thing if the cyclone tracks are similar but with differing moisture flux as they would still be able to test the sensitivity of precipitation totals to moisture flux. Therefore, I don't think it is sensible to examine the suitability of the ensemble by looking at the spread in precipitation as is done in table 2.
9. Page 7, line 16: Can the authors be more specific about the section containing the full evaluation. Currently they say is it 'below', below where?
10. Page 10, lines 11-12: There is no motivation given for testing sensitivity to the convection, microphysics schemes or nesting in WRF. What was the reason for performing these simulations? Did the authors have a hypothesis that they wanted to test? What are the differences between the schemes?
11. Page 10, line 27: The authors claim that there is a systematic improvement with decreasing lead time. However, this is difficult to detect in the spatial verification statistics shown in table 2.
12. Page 11, line 4: The authors conclude that there is no difference in performance when changing the cumulus scheme or nesting but do not analyse this result. Did they expect to see a difference? If so, why are the results insensitive to these choices? What is the conclusion of this experiment?
13. Page 11, line 28: Here the authors present figures 4d-f and 4g-I but do not analyse these figures. If they are not referred to in the text should they be removed?
14. Figure 3: The right-hand edge of the figure has been cut off. It is also not clear what cross-section figures 3i-k are for. Could the cross-section be added to figures 3f-h respectively?
15. Page 14, line 18 and elsewhere: The authors conclude that the ensemble spread becomes increasingly larger when going back in time. Although this is an intuitive result, it is not possible to conclude this from 3 points only. More case studies would be needed to confirm this.

16. Page 14, lines 12-22:  The authors do not refer to any figures in this analysis section. Which figures are used?  Is this where the analysis of 4d-f and 4g-l is performed?

17. Figure 4:  Why is a different domain used in figures d-f?  Is the ensemble track position agreement in the North-Atlantic relevant?  It appears as though the track agreement over Switzerland is similar for all 3 cases, is this correct?

18. Figure 4:  I do not know what figures 4g-i are showing.  Please explain these figures in the text.

19. Figure 4: These are quite complex figures.  Are the country outlines important?  Perhaps they could be removed?  Or only Switzerland included?

20. Page 16, line 18:  The authors say that the model 'agrees' with the CombiPrecip precipitation.  How did they come to this conclusion?  The time evolution of the CombiPrecip appears to lie outside the ensemble spread for a large part of the timeseries implying poor agreement.

21. Page 18, line 13:  Why is the fact that the storm track for max precipitation in 20CR and downscaled simulations is similar 'remarkable'?  Did the authors expect to see large differences in the position of the storm track?  Doesn't the similarity indicate that the track of the cyclone is the primary control on precipitation accumulations over Switzerland?

22. Figures 7, 8 and 9:  These figures are of very poor quality.  They do not contain lat/lon, a colour bar or continent outlines.  This makes the analysis impossible to follow.

23. Page 19, lines 5-15:  Analysis of figures 7-9 is described in just 13 short lines.  Is it therefore justified to include all 18 figure sub-panels?

24. Figure 10: As far as I can tell both the colours and size of dots represent the precipitation intensity.  Are both methods needed?

25. Page 20, lines 8-13:  These lines describe figure 10.  This is a complex diagram and the analysis of it is rather brief (6 lines).  Given that one of the major objectives of the paper is to examine 'sensitivity to cyclonic moisture flux' (title) the analysis should be expanded.

26. Page 21, line 12:  The authors describe the act that one of the ensemble members produces higher precipitation for the 1910 event than those observed as 'remarkable'.  I'm not sure why this is remarkable.  The purpose of the ensemble is to represent the range of plausible situations given the large-scale flow conditions so if all of the ensemble members underpredicted the observed precipitation totals then this would be a poor ensemble.  Perhaps I have misunderstood something here?

27. Page 23, lines 1-8:  While the discussion of PV streamers is interesting, it is not a result of this paper, so it should not be in the results section.

28. Page 23, line 19:  Whether short spin-up periods are 'preferable over long spin-up periods' depends on what you are trying to optimise and is not a general result. I think the objective in this study was to minimise spread in the ensemble tracks so as test sensitivity of precipitation to moisture flux rather than track position.  Another objective may well have resulted in a different optimal spin-up period.

29. Page 23, line 20: What are slow-reacting features?

30. Page 23, line 20: Again 'good results' depends on what you are trying to achieve. Small differences in the ensemble will occur if the cyclones are already present in the outermost model domain.  Is that the point?

31. Page 24, line 7:  I do not think you can conclude that uncertainty increases gradually when going back in time using 3 case studies only.

32. Page 24, line 9:  Similarly, concluding that dynamical downscaling is less accurate going back in time is difficult using 3 case studies only.  There are many other factors that would increase the uncertainty for specific case studies.

33. Page 24, lines 14-23:  This is an excellent summary and it would be nice to see a more in-depth analysis in the main body of the text to support these conclusions.

34. Page 24, line 26: How do you conclude that the 20CR tracks are not 'realistically located' for the 1876 case?  Are you stating that 20CR produces unrealistic tracks, or simply that the uncertainty in the position of the track is large for this case potentially because it is a complex situation?

---

## Referee Comment (RC3) · Anonymous Referee #3 · 11 Aug 2019

Review of nhess-2019-174:

"Simulations of the 2005, 1910 and 1876 Vb cyclones over the Alps – Sensitivity to
model physics and cyclonic moisture flux"
by
Peter Stucki, Paul Froidevaux, Marcelo Zamuriano, Francesco Alessandro Isotta, Martina
Messmer, Andrey Martynov

**Recommendation: major revisions**

The authors present three cases of extreme precipitation along the Alps associated with
the classical synoptic situation Vb. They utilize downscaling of different reanalysis
products for their analysis, where they employ WRF in a nested method to go down to 3
km resolution in the innermost domain. The authors test several sensitivities of their
results with respect to the simulation setup, addressing both numerical as well as physical
changes. They find that the lead time is among the most crucial parameters. While I find
the manuscript well written, I struggle to see a clear motivation and conclusion for the
study. The motivation is probably also somewhat difficult, because the manuscript tries to
address many different questions at the same time: (1) What is the best downscaling setup
for the cases in question. (2) What lead to the extreme precipitation. (3) Description of
the three individual cases and comparison. The treatment of all these topics makes the
paper sometimes difficult to follow. Regarding the conclusions, similar arguments apply
and the authors sometimes appear to state opinions/speculations that are not necessarily
solidly grounded in the material they presented. However, after responding to some of
my major concerns, I believe this manuscript is acceptable for publication in NHESS.

**General Comments:**

In the title, I am not sure what the authors really mean with the "cyclonic moisture flux".
Is it a moisture flux going in a cyclonic direction or the moisture flux associated with a
cyclone? Given this ambiguity, I encourage the authors to further clarify this aspect in
their title as well as throughout the manuscript to help the reader making a clear link.

Given that there are only three cases, and the fact that a majority of the reasoning for the
downscaling is based on the most recent case, the authors should be more cautious about
general statements on downscaling procedures, as the results are highly sensitive to the
case(s) at hand. While the technicalities that were overcome by the group are certainly
impressive, it is still not clear to me how generic these results can be treated. In order to
make a more general claim about the downscaling for Vb situation, one would need to
explore many more cases to arrive at a firm conclusion. The authors should thus make it
clear that this study can at most give an indication what one might need to test in order to
arrive at a more general conclusion.

What made the authors pick a 10-day spin-up time? It seems excessively long for the
investigation of such a regional and meso-scale influenced precipitation event. At the
end, the authors arrive at a 1-day spin-up time anyway, but the vastness of the parameter
space is not sufficiently motivated, similar to some of the other sensitivity tests.

The list of sensitivities is extremely exhaustive, ranging from resolution to resolution
ratios over spin-up time to parameterizations and model domains. The enormous
parameter space is rather difficult to grasp and all results will primarily be in relation to

the 2005 case, with general deductions being rather limited due to the specifics of the case. In general, it would aid the reader if the authors more clearly state their working hypotheses as well as the reasoning for their choices and expectations. This will make it more straight forward to follow the ensuing arguments.

The authors often refer to reproducing "correct" precipitation amounts. What is meant by correct? Presumably compared to observations, though the authors list several observations that are used. In addition, all of these "observations" also rely on some sort of downscaling and gridding, as data voids need to be filled. The authors, however, do not provide a detailed analysis of the representativeness of these observations. They refer to other studies that addressed these to some extent, but given the specifics of the case studies, the authors should also comment on the validity of the observations before comparing the model simulations to the data in order to claim "correctness".

For the validation of, for example, precipitation, it has proved useful to use feature-based detections that consider location, shape, and timing. Why have the authors not considered more such verification tools for the study at hand? It appears the method referred to as EMD is in fact such a measure, though it appears confusing why the authors use a visual inspection for a quantitative comparison. The reasoning for the choices and omission of other tools should be clearly motivated.

For the philosophical concluding paragraph on page 25 not much hard evidence has been provided in the manuscript for the claims put forward. It thus reads more like a written piece of opinion than a well and quantitatively justified conclusions.

**Specific Comments:**
The page (P) and line (L) numbers refer to the ones in the manuscript.

P1 L23: "to the cyclonic" and see comment above about the ambiguity of "cyclonic moisture flux".

P1 L28: "accurate directions" with respect to what? What is the reference?

P8 L1: The precipitation data is interpolated. Can the authors please clarify if the interpolation was carried out in such a way that the total precipitation was unaffected by the interpolation? Depending on the kind of interpolation, the results for the totals can deviate.

P11 L12: The authors speculate on the differences between ERA-Interim and 20CR in terms of moisture distribution, though the authors could provide direct evidence for their claim by investigating differences between ERA-Interim and 20CR fields for the case at hand in more detail.

P11 L22: Why did the authors chose to identify and track cyclones using geopotential at 500 hPa? This seems rather unconventional and needs further motivation.

Fig. 4: The cyclone tracks for ERA-Interim look very edgy. In order to compare them better to the other plots, a grid that is not fixed to the grid spacing of the data could be beneficial, which is most often done in other cyclone track algorithms, see also comment above about cyclone track determination in this manuscript.

Fig. 7, 8, and 9: I find these figures not very legible. Maybe this is due to the downgrading of the figure quality for the review process, but otherwise the readability of the information of these figures needs to be significantly improved. In particular the arrows are not very visible.

P22 L21: The authors should explain how PV is produced in the downscaling process, as this appears to be crucial in their arguments.

P23 L24: How can the authors conclude that "nudging smaller domains can still be beneficial"? Has any evidence been provided in this study to support such a claim?

P24 L1: The authors should be more specific what they are referring to with "traditional spatial verification scores".

---

## Author Comment (AC1) · 27 Sep 2019

General comments:

The authors downscale three historic flood events caused by so-called Vb-cyclones and analyse the conditions under which the high-resolution simulations capture precip- itation over Switzerland best. They compare model set-ups with different initialization periods, parametrizations and nudging. The most important factor determining the result with respect to precipitation is the correct representation of the cyclone path.

The article is well written and structured. It addresses three extreme precipitation events and is therefore within the scope of the journal. The authors systematically analyse different model configurations and initial data sets which allows them to identify influencing factors of higher and lower importance. The result is a good reference for other scientists applying dynamical downscaling to study extreme precipitation events. It also shows the pros and cons of using 20CR reanalysis for initializing regional model simulations, as it turns out that deviations to observations increase for events lying further in the past.

There is however one aspect that is missing in the paper. The authors downscale to a precipitation resolving resolution of 3 km. As a motivation they cite Zängl 2007 and Prein et al. 2015 who analysed the 2005 event (page 3, line 29). The simulations performed by Stucki et al. would allow to study the effect of convection resolving simu- lations versus simulations with convection parametrization also for the two other cases. The paper would strongly benefit from adding an additional section which discusses the effect of the last downscaling step in more detail.

**We thank the reviewer for the clear and helpful comments. Specifically, we appreciate the valuable note suggesting analyses on the effect of the last downscaling step and the associated convection. This is indeed an important aspect that we have neglected in the previous version. To address the suggestion, we have now redrawn Figure 5 and added two panel rows showing the contribution of convective precipitation in the 9-km domain, where precipitation is parameterized. This gives insights in the process of (embedded) convection during the events. In addition, we add a new Figure 11. It shows surface weather simulated for a historical maximum-precipitation member at three instances in time. With this, we can better show how shifts in the cyclonic flow and moisture transport translate into regional to local surface weather and precipitation patterns.**

Specific comments:

p.2 l.4: The cited article (BAFU 2005) is missing in reference section.

**We thank the reviewer for the hint. It is in fact Bezzola et al. 2008, which is available in the reference list.**

p.2 l.12: Cyclogenesis of Vb cyclones can be outside the Mediterranean region but most of them pass the Golf-of-Genoa region (e.g. Messmer et al., 2015).

**We have adopted the suggested wording, and we agree that a broader definition is more adequate. It reads now as follows: "Cyclones on a Vb track are associated with heavy to extreme** precipitation over Central Europe, and particularly north of the Alps (Hofstätter et al., 2016, 2018;

Nissen et al., 2013). Most of the cyclones following the Vb trajectory are generated in the western Mediterranean region, and most of them pass the Golf-of-Genoa region (Hofstätter and Blöschl, 2019; Messmer et al., 2015). For this, they are also called Genoa Lows at this stage; Bezzola et al. 2008)**"**

Table 2: Does a positive mean absolute error means that the model initialized with 20CR produces more precipitation than observed and that initialization with ERA40 (p.11) overestimates precipitation even further? This is probably not the case as it would be in contrast to Fig. 2. Please define what a positive MAE value indicates. From looking at Fig. 2 I also don't understand how MAE48h can be 27.75 for sp10 when averaging over 10 ensemble members. Shouldn't the error be much higher?

**We have added a short explanation of MAE in Sect. 3.2. MAE gives absolute values, that is, the deviation or distance from the observation – regardless of its sign. The bias is actually negative. We also checked the calculations and come to the same results.**

**The new text reads as follows: "**The MAE measures the average distance between forecast and observation, and is preferred over RMSE because it is more resistant to outliers, and over correlation coefficients because we are more interested in accuracy than linear association (Joliffe and Stephenson, 2012).**"**

p.11 l.28: Please give a short explanation how cyclone fields are calculated (1-2 sen- tences). To understand what is shown one shouldn't need to read another paper. Please also clarify if cyclone fields and cyclone tracks are both calculated at SLP.

**We agree that this needs to be clarified. We have inserted a short explanation, and we have adapted the caption in Figure 4. The new text reads as follows: "**The algorithm detects cyclone fields in terms of a finite area around a regional SLP minimum, that is, by a closed SLP contour line. The regional SLP minima for each cyclone life cycle are stored as cyclone tracks, and the presence or absence of a cyclone is represented in a binary field for each grid point and time step.**"**

p.12 l.17 +others: You use storm track and cyclone track synonymously. This is con- fusing as you only show cyclone tracks. The term storm track is mostly used for the variance at synoptic time scales (e.g. Hoskin and Hodges, 2002: New perspectives on the Northern Hemisphere winter storm track. Journal of Atmospheric Sciences, vol. 59, Issue 6, pp.1041-1061). I suggest that you change "storm track" to "cyclone track" everywhere in the manuscript.

**We fully agree; the suggestion has been adopted.**

Fig.4 first row: Genoa cyclogenesis is not visible on these figures.

**We agree. Shown are cyclone tracks at the 500-hPa level in the 2° by 2° reanalysis. We have made this clear in the caption.**

Fig.4 middle row: There is a time step indicated in each panel but the caption suggests that different time steps are combined. This is confusing.

**We have rephrased the caption and we emphasize now that it summarizes the ensemble at one specific time step. The caption reads as follows: "**. (g,h,i) Cyclone centers identified for sea level pressure at (g) 21 August 2005 18 UTC, (h) 13 June 1910 18 UTC, and (i) 10 June 1876 18 UTC, the same time steps as in (d, e, f). Colored dots and the tick marks in the color key indicate the

number of cyclone tracks located at a specific grid point at the respective time steps. Greyscaled lines mark the cyclone tracks over the period of 48 h before to 48 h past the respective time steps. Darker (lighter) grey shades indicate more (less) cyclone tracks along a certain path**"**

Fig.4 last row: Please redraw. It is almost impossible to see the cyclone tracks. In ad- dition, the caption states "as in middle panels" even though g, h and I show a sequence of days while the d, e and f show single time steps. I also don't understand why the numbers on the colour bars differ for g, h and i.
**The panel is redrawn, with larger and darker lines, a smaller map area and larger circles. In addition, we have rephrased the caption. We also describe the last panel better in the text now, which in fact supports our conclusions. This is also in line with a suggestion by Reviewer 2.**

Fig. 5a: There is a very prominent precipitation peak before the event starts. The text does not give any explanation for this peak.
**This is true. We have now addressed the peaks in the text, including that they are convection-driven. We have also added the simulation results from WRF-ERA-Interim. We cannot fully conclude on the reasons for the overestimation during the spin-up, but found that at the very beginning of the model initialization, the interpolated temperature field is coarser in 20CR-WRF for the 9-km domain, which leads to areas with high temperatures over the Alps and excess moisture production in the first hours of the simulation. This mechanism is now summarized in the text.**

Figs 7, 8 and 9: Are the black dots supposed to indicate the land sea mask? This does not come out neither in print nor on my screen. Please redraw.
**This is a common issue for all reviewers. Apparently, this happened during rasterization of the original vector format to PNG format, such that it can be inserted in the submitted Word document. We still have trouble to produce a good PNG file, but hope that the image quality is now sufficient. The PDF to be submitted for print production should be in good quality.**

p.24 last paragraph: The decreasing differences to observations with time in 20CR are probably due to the increasing quality/amount of data that is assimilated in 20CR. It would be good to mention how the input data for the 20CR product has changes between 1976 and 1910.
**This is a good suggestion. We inserted the according numbers from the ISPD database in the conclusion; they are really quite illustrative.**

Fig. A2 caption: "(time periods as in Figure 2)". Fig. 2 only shows 2005 Fig. A2 shows all 3 episodes.
**We thank the reviewer for this remark; it should refer to Fig. A1, of course.**

Technical corrections:
p.9 l.11: Value in the table is 0.59 the value in the table captions is 0.49 (sp10 48-hour precip). Which one is correct?
**We thank the reviewer for this hint. The value in the table is correct; we have corrected the one in the text.**

**Further changes made**
- **Additional references: Hofstätter and Blöschl, 2019; Zbinden, 2005 (Annals of MeteoSwiss); Cioni and Hohenegger, 2019; Coppola et al., 2018; Compo et al, 2015; Joliffe and Stephenson, 2012; Wernli et al., 2008.**
- **Figure A2 redrawn with the maximum- and minimum-precipitation members. This makes more sense since we focus on these in the text.**

---

## Author Comment (AC2) · 27 Sep 2019

Simulations of the 2005, 1910 and 1876 Vb cyclones over the Alps – Sensitivity to
model physics and cyclonic moisture flux
Peter Stucki
et al. nhess-
2019-174

There are 3 main objectives of this paper (i) to find a setup of the WRF model that is
adequate for dynamical downscaling from 20CR, (ii) to investigate sensitivity of heavy
precipitation to cyclonic moisture flux and (iii) to assess the uncertainty along the
downscaling steps and among the ensemble members for historical cases.  The paper is
well written but I'm concerned about the balance of the paper.  The paper is dominated by
the technical aspects of performing downscaling for historical events (with poorly-defined
motivation for performing model tests and testing ensemble suitability) and contains too
brief analysis of the cyclonic moisture flux to achieve objective (ii) (see general
comments below).  If the points below are addressed this paper would be suitable for
publication in NHESS.

**We thank the reviewer for pointing to some important aspects and potential flaws of
the manuscript. We think that while addressing the concerns, we provide now a
more balanced manuscript regarding the technical and dynamical parts of the
article. The motivation, or rather the rationale, of doing the series of experiments is
now described in a more explicit and clear way. Considering also comments by
Reviewer 1, we have better defined and extended the analyses of cyclonic moisture
flux with new figure panels, a new figure and extended text.**
**In all, we are grateful for the review, which supported us in improving the quality of
the manuscript regarding some crucial aspects and clarity.**

General comments
1. I'm concerned that there is no motivation given for testing sensitivity to the
   convection, microphysics schemes or nesting in WRF.  What was the reason for
   performing these simulations?  Did the authors have a hypothesis that they wanted
   to test?  What are the differences between the schemes?  The authors conclude that
   there is no difference in performance when changing the cumulus scheme or
   nesting but do not analyse this result.  Did they expect to see a difference?  If so,
   why are the results insensitive to these choices?  What is the conclusion of these
   experiments
   and how general are they, i.e. would the same hold for other historical cases or
   are they specific to these cases? If the conclusions are case specific, then
   perhaps this analysis could be reported in an appendix?

   **We see the need for a better reasoning about the experiments. To begin
   with, we state more clearly why we started with ample spin-up time, then
   why are not satisfied with the standard setup, and then derive why we go
   with decreasing spin-up. Regarding the specific question of 'sensitivity' to a
   number of schemes and settings, we now stat more clearly that "The goal of
   these last experiments is not to achieve a thorough sensitivity assessment for
   each tuning option, but to make sure that we have not chosen a sub-optimal
   setup. This is checked by further modifying a number of configurations of the
   WRF model which may have an influence on the simulation performance
   according to literature and from our experience." In turn, this means that we**

**do not aim to analyze each tuning option in detail.**
**Some of this concern could also stem from the fact that we use the term 'sensitivity' in the title, which may provoke expectations of a thorough testing of many schemes and options. In a narrow sense, the term might only be acceptable for the more detailed tests regarding the spin-up. We use 'sensitivity' in a wider sense, where we also allude to checking the other tuning options in less detail as well as to the dynamic sensitivities addressed by contrasting members, for instance. In fact, we have considered replacing the term 'sensitivity' in the title and elsewhere (e.g. Implications from …, the role of …, etc.). We have not found a concise alternative term that would include all these meanings, and we think that the meaning in this article should become clear when reading the abstract.**

2. There is some confusion in the paper over what constitutes a 'good' ensemble spread. The answer to this depends on the hypothesis being tested. At some points in the paper the authors claim the ensemble is good or bad by examining spread in the precipitation totals (figure 2, table 2) concluding that the 10-day forecast runs 'too freely' because the precipitation accumulation spread is large. However, later in the paper they examine the spread in cyclone tracks (figure 4) and conclude that there is a 'good' spread in storm track position for the 2005 and 1910 cases but not the 1876 case (meaning smaller track position differences in the ensemble). If the focus of the paper is to test the sensitivity of precipitation accumulation over Switzerland to cyclonic moisture flux, some spread in precipitation accumulation is surely necessary? However, spread in precipitation that occurs due to factors such as cyclone position presumably need to be minimised? Is this the rational for the later measure of ensemble suitability? If so, why is precipitation spread used in the early analysis of ensemble spread? There appears to be some inconsistency in the analysis of ensemble suitability in the paper which needs to be clarified.

[Figure]

**We agree that we need to address in more detail what we mean by a 'good spread', and the comment has helped a lot in this respect. We discussed the trade-off between 'plausible runs' and 'necessary spread'. In the end, we think that we can justify our line of thoughts well in an additional paragraph in Sect. 3.1.**
**In addition, we include a reference (Coppola et al., 2018, https://doi.org/10.1007/s00382-018-4521-8) to a thorough test of multi-model ensembles, where the inclined reader can actually look up how state-of-the-art models perform in a comparable Alpine setting, see their Figure 5 to the right.**
**The new text reads now as follows: "**In all, we cannot be satisfied with these results yet. On the one hand, we aim to investigate particularly flood-inducing features of Vb-cyclones. For this, a certain variability in the ensemble is helpful and necessary. For example, we can find and assess (non-)decisive features by means of opposing ensemble members. On the other hand, we also need to ensure that our downscaling experiment delivers plausible results, especially regarding precipitation intensities and patterns. For this, the deviations from the observations must not become too large in the ensemble. A somewhat smaller spread of the simulated precipitation for the 2005 case would also increase our confidence that

the simulation of historical events will produce reasonable and valid results.
In short, we would have expected less underestimation and smaller deviations with
this downscaling configuration (cf. Coppola et al., 2018, their figure 5, for
estimations of accumulated precipitation over the Alps from a multi-model
ensemble)."

3. Analysis relating to the sensitivity of precipitation to cyclonic moisture fluxes
   (figures 7-10) is described in just 19 lines. This is rather brief for 4 figures,
   especially given that one of the major objectives of the paper is to examine
   'sensitivity to cyclonic moisture flux' (title). The analysis must be expanded to
   provide a better balance
between the technical aspects and the scientific hypothesis testing analysis and to
achieve objective (ii).

   **We agree on this comment, which is similar to concerns by Reviewer 1, and it
   has made us review and revise the manuscript in several ways.**
   **The discussion of Figure 10 is actually done later in the text (see also reply to
   specific comment 25). We have introduced more references to Figure 10, for
   instance, to enhance the context of our findings.**
   **To be more balanced between technical and dynamic aspects of sensitivity, and
   to address the innermost domains better (see comments by Reviewer 1), we
   have now redrawn Figure 5 and added two panel rows showing the
   contribution of convective precipitation in the 9-km domain, where
   precipitation is parameterized. In addition, we add a new Figure 11. It shows
   surface weather simulated for a historical maximum-precipitation member at
   three instances in time. With this, we can better show how shifts in the
   cyclonic flow and moisture transport translate into regional to local surface
   weather and precipitation patterns. An according clause is added in the
   Abstract.**

Specific comments
   1. Page 1, line 22: What is 'moderate' spectral nudging?
      **Moderate refers to wavelengths of 1500 km; this is only defined in the text. We
      see that the term is not commonly used and decided to remove the information
      from the abstract, and instead focus on the most robust result, the
      initialization period. This is also in accordance to the comment by Reviewer 3.**

   2. Page 2, line 20: How is the moist air 'let' around the Eastern Alps? Do you mean
      advected?
      **The sentence is rephrased in active voice with 'flowed over and around the
      Eastern Alps'. Note that in agreement with Reviewer 3, the dynamics of
      'cyclonic moisture flux' is now better defined here.**

   3. Page 4, line 28: The authors state that they use a 'consistent part of the
      calibration period, which is accordingly slightly reduced'. I'm unclear what the
      consistent part of the calibration refers to. Please could the authors expand on
      this?
      **We have modified the sentence to improve clarity. The stations used are the
      one that were available in the days of interest in 1910 or 1876 and at the**

**same time during most of the calibration period. We repeat the calibration period (1981- 2010, given also two sentences above). The calibration period is slightly reduced to the days when all chosen stations were available (the method needs to have the same set of stations in the calibration period, in our case from 1981-2010, and the reconstructed period, i.e. the days in 1910 or 1876).**

4. Page 5, line 5. It would be useful to know if any of the assimilated surface pressure observations were located in Switzerland.
   **We thank the reviewer for the hint. We have introduced the numbers here, and come back to the total number of assimilated stations in the conclusions. This is also in line with a suggestion by Reviewer 1.**

5. Figure 1: The numbers on the colour bar have been cut off.
   **We thank the reviewer for the remark. We have replaced Figure 1.**

6. Page 6, line 26 and page 16, line 18: The authors refer to 'two peak episodes' but in figures 2 and 5 the CombiPrecip dataset does not show 2 peak episodes. Instead there is continuous high precipitation rates over a 30hr period.
   **We were taking the perspective from the simulation, but agree that this is a misleading phrasing. We rephrased on Page 6 to 'Furthermore, CombiPrecip shows high precipitation rates over most of the analyzed period. In contrast, the downscaled ensemble has only two periods of very high precipitation rates (at around 12 and 36 hours after initialization), and obviously, there is too little precipitation between these two simulated peak episodes'. A similar phrasing is employed on Page 16. We think that it fits also better with the revisions suggested by Reviewer 3.**

7. Figure 2: The right-hand axis does not have any units. Also, it is not clear what the red numbers represent.
   **We thank the reviewer for pointing this out. We have redrawn Figure 2, and have added a # to mark the quartile members, and have also added it in the caption for clarity.**

8. Page 7, line 1: It is not surprising that a 10-day forecast exhibits large spread in the ensemble. However, this is not necessarily a bad thing if the cyclone tracks are similar but with differing moisture flux as they would still be able to test the sensitivity of precipitation totals to moisture flux. Therefore, I don't think it is sensible to examine the suitability of the ensemble by looking at the spread in precipitation as is done in table 2.
   **We suppose Figure 2 is meant instead of Table 2. This comment helped a lot for clarifying our rationale for the downscaling, and the follow-up series of experiments. Following also the comment on this topic by Reviewer 3, we now explain this in more detail in the analysis and interpretation of Figure 2. Later in the text, we also explain why we are not satisfied with the results and how we came up with the follow-up experiments. See also our reply to comment 10.**

9. Page 7, line 16: Can the authors be more specific about the section containing the full evaluation. Currently they say is it 'below', below where?
**We have introduced 'Sect. 3.2'.**

10. Page 10, lines 11-12: There is no motivation given for testing sensitivity to the convection, microphysics schemes or nesting in WRF. What was the reason for performing these simulations? Did the authors have a hypothesis that they wanted to test? What are the differences between the schemes?
**We suppose it is Page 8 instead of 10. There, we now give the reasoning for the last series of experiments: '**The goal of these last experiments is not to achieve a thorough sensitivity assessment for each tuning option, but to make sure that we have not chosen a sub-optimal setup. This is checked by further modifying a number of configurations of the WRF model which may have an influence on the simulation performance according to literature and from our experience.**' We thank the reviewer for this remark; we think we are much clearer now about the purposes of the three series of experiments. See also our response to Reviewer 1 on this issue.**

11. Page 10, line 27: The authors claim that there is a systematic improvement with decreasing lead time. However, this is difficult to detect in the spatial verification statistics shown in table 2.
**We see that detection by eye might be difficult. Therefore, we have included the calculated Theil-Sen slopes in the text for better documentation. The new text reads now as follows: "**Theil-Sen slope estimates are calculated over sp10, sp7, sp5, sp3, and sp1 for all measures; they are all negative (MAE 24 h: -0.2; MAE 48 h: -0.70; BOX 24 h: -0.03; BOX 48 h: -0.03; VIS 24 h: -0.75; VIS 48 h: -1.13; EMD 24 h: -18.5; EMD 48 h: -17.9). Although the trends are not significant in Mann-Kendall tests (or not clearly attributable, due to the small sample), the negative slopes indicate that performance generally increases with decreasing spin-up time**"**

12. Page 11, line 4: The authors conclude that there is no difference in performance when changing the cumulus scheme or nesting but do not analyse this result. Did they expect to see a difference? If so, why are the results insensitive to these choices? What is the conclusion of this experiment?
**We agree that the explanations are not very detailed in the manuscript. This is mainly because, as we state in the text now, the goal of these last experiments is not to achieve a robust sensitivity assessment for each tuning option, but to make sure that we have not chosen a sub-optimal configuration of the WRF model. We have inserted some explanations of why a different setup might change the precipitation pattern and indicate some more literature. Some of our reasoning is given below in more detail.**
**The microphysics parameterization describes the hydrometeors (cloud water, raindrops, ice, snow, graupel, etc.), the number concentration, size, fall speed etc. These descriptions differ between the chosen parameterizations. These differences play a role in cloud development and hence, also in precipitation amounts and patterns. As the microphysics parameterization is responsible for all the precipitation in the innermost domain (no cumulus parameterization), changes in this parameterization can certainly affect precipitation amounts and patterns. Compared to**

**Thompson, the ice particles (ice, graupel and hail) are described in less detail in the Ferrier parameterization. The reason, why the difference is rather small might be that we only look at summer events, where ice particles are less important. We have included a sentence about this argumentation in the manuscript.**

**The two cumulus parameterizations that have been used show differences in scale-dependence. The Kain-Fritsch scheme is one of the most commonly used parameterizations over Europe. The scale-dependent scheme is designed to improve the realizations in the so-called grey zone between 10 and 5 km.**

**Since we do not employ a cumulus parameterization in the inner nest, the changes in the outer domains seem to be relatively small, when changing the cumulus scheme. This might be because the description of the convection in this case is not triggered by fine scale structures, but mainly by lifting along the orography. We have included a short description of the differences in the two cumulus parameterizations.**

**Two-way nesting can be helpful, if an event can be modified by the small-scale features that are resolved in the innermost domain. In such a case, the result of the innermost domain can be transported to the coarser domains as well. In a Vb event one or two-way nesting might not make a big difference, as the cyclone is steered by large-scale atmospheric flow.**

13. Page 11, line 28: Here the authors present figures 4d-f and 4g-I but do not analyse these figures. If they are not referred to in the text should they be removed?

    **The references can be found on p11 bottom and p12 top, then again on p12l18 for Figure 4a, d and g, on p12l26 for Figure 4b, e and h, and on p14L8 for Figure 4c, f and i. We see that more references could be helpful, though, and introduced some more in the text where appropriate.**

14. Figure 3: The right-hand edge of the figure has been cut off. It is also not clear what cross-section figures 3i-k are for. Could the cross-section be added to figures 3f-h respectively?

    **We do not see the cut in our submission. However, we have redrawn Figure 3, adding all cross-sections to simplify referencing, and have rephrased the caption for more clarity.**

15. Page 14, line 18 and elsewhere: The authors conclude that the ensemble spread becomes increasingly larger when going back in time. Although this is an intuitive result, it is not possible to conclude this from 3 points only. More case studies would be needed to confirm this.

    **We did not intend to make a general claim here. The whole paragraph is meant as a summary / discussion of the synoptic analyses before we go on. This is now clearly stated at the top of the paragraph.**

16. Page 14, lines 12-22: The authors do not refer to any figures in this analysis section. Which figures are used? Is this where the analysis of 4d-f and 4g-l is performed?

    **See also the reply to comment 14. We include now more specific references to the respective Figures, and add some more text to guide the reader.**

17. Figure 4: Why is a different domain used in figures d-f? Is the ensemble track position agreement in the North-Atlantic relevant? It appears as though the track agreement over Switzerland is similar for all 3 cases, is this correct?
**The larger area is chosen to show the relatively good performance of 20CR in the region of interest in comparison to other regions in this area. We have extended the phrasing to '**Overall, the analyses at synoptic scales (Figures 3 and 4) show that differences among the 20CR members are substantially smaller over the region of interest (Southern and Central Europe) than over other regions of the North Atlantic / European sector (Figure 4 d, e and f); this corresponds to the relatively high density of assimilated stations over Central Europe (not shown; see Compo et al. 2015).**'**

18. Figure 4: I do not know what figures 4g-i are showing. Please explain these figures in the text.
**The Figure panels have been redrawn, according to a comment by Reviewer 1, and the caption has been extended. As requested, we now analyze the three panels in more detail. We thank the reviewer for this comment; we think the revisions make our points much clearer now.**

19. Figure 4: These are quite complex figures. Are the country outlines important? Perhaps they could be removed? Or only Switzerland included?
**We would like to keep the country borders as is because they can be useful for localizing the features. We used some lighter grey shades where appropriate, though.**

20. Page 16, line 18: The authors say that the model 'agrees' with the CombiPrecip precipitation. How did they come to this conclusion? The time evolution of the CombiPrecip appears to lie outside the ensemble spread for a large part of the timeseries implying poor agreement.
**Please see the reply to comment 6.**

21. Page 18, line 13: Why is the fact that the storm track for max precipitation in 20CR and downscaled simulations is similar 'remarkable'? Did the authors expect to see large differences in the position of the storm track? Doesn't the similarity indicate that the track of the cyclone is the primary control on precipitation accumulations over Switzerland?
**We have removed the word in the revised manuscript.**

22. Figures 7, 8 and 9: These figures are of very poor quality. They do not contain lat/lon, a colour bar or continent outlines. This makes the analysis impossible to follow.
**Similar comments are made by all reviewers. We think that the loss of quality was introduced when converting from vector format to PNG, and during the upload process. In the revised version, we have a better resolution of the PNG file, and readability of the PDF for print is good in our view. Note also that the color key had been missing.**

23. Page 19, lines 5-15: Analysis of figures 7-9 is described in just 13 short lines. Is it therefore justified to include all 18 figure sub-panels?

**This is an eye-opening comment. We have realized that the analyses needs more details, and we have extended the paragraph accordingly. We think that it links also much better with the additional analyses suggested by Reviewer 1.**

24. Figure 10: As far as I can tell both the colours and size of dots represent the precipitation intensity. Are both methods needed?
**Yes, we prefer to keep both elements. The co-authors found in the internal review that comprehension of the Figure content is easier.**

25. Page 20, lines 8-13: These lines describe figure 10. This is a complex diagram and the analysis of it is rather brief (6 lines). Given that one of the major objectives of the paper is to examine 'sensitivity to cyclonic moisture flux' (title) the analysis should be expanded.
**We agree that the text is rather short at this point. In fact, we come back to Figure 10 in a later discussion paragraph. Following the suggestions also earlier in the review, we introduced a specific reference. Note that according to the comment by Reviewer 1, the analysis of 'cyclonic moisture flux' in the two innermost model domains is extended, including a new Figure 11.**

26. Page 21, line 12: The authors describe the act that one of the ensemble members produces higher precipitation for the 1910 event than those observed as 'remarkable'. I'm not sure why this is remarkable. The purpose of the ensemble is to represent the range of plausible situations given the large-scale flow conditions so if all of the ensemble members underpredicted the observed precipitation totals then this would be a poor ensemble. Perhaps I have misunderstood something here?
**We can follow this argumentation. Hence, the word 'remarkable' may not be appropriate here, and we omit it in the revised version of the manuscript.**

27. Page 23, lines 1-8: While the discussion of PV streamers is interesting, it is not a result of this paper, so it should not be in the results section.
**Throughout the article, discussion and interpretation follow the presentation of results. However, we have rephrased the discussion without PV streamers. In accordance to a similar comment by Reviewer 3, we have rephrased the discussion without PV streamers. The aim of the paragraph is to mention the underlying physical processes and interaction between the Alpine orography and upper-level troughs. These physical processes can be referred to without introducing the notion of PV, which is indeed not introduced in our study. In fact, this has allowed to focus more on explaining the untypical results from the 1876 case.**
**The new text reads as follows: "**This behavior of the model can be explained by the interaction of the Alpine orography with atmospheric circulation. Indeed, Vb cyclone trajectories are typically initiated by deepening upper-level troughs, which finally cut off from the westerly flow when passing over the Alps (e.g. Awan and Formayer, 2017). The interaction of upper-level troughs with the Alpine orography have been described in detail (Buzzi and Tibaldi 1978; Aebischer and Schär 1998; Kljun et al. 2001); the underlying processes include flow splitting and lee cyclogenesis, with further amplifications of the

cyclone formation by frontal retardation and latent heat release due to orographic lifting. The combination of these processes implies that the cyclones are formed on the lee of the right side of the Alps, typically over the Ligurian Sea. In 20CR, however, the Alpine orography is very coarse, smoothed and reaches only about 1000 m a.s.l. (cf. Stucki et al., 2012). Hence, the influence of the Alps on the large-scale flow is limited in 20CR. Given also that the 1876 case is least confined by pressure observations, this allows untypical cyclone tracks in many 20CR members. Once accounting for a more and more realistic orography throughout the downscaling steps with WRF, the high-resolution runs may thus end up in a compromise simulation - driven both by the WRF model physics and by the 20CR input flow. In other terms, the large-scale flow forced from 20CR might not be compatible with the orography of the high-resolution domains**."**

28. Page 23, line 19: Whether short spin-up periods are 'preferable over long spin-up periods' depends on what you are trying to optimise and is not a general result. I think the objective in this study was to minimise spread in the ensemble tracks so as test sensitivity of precipitation to moisture flux rather than track position. Another objective may well have resulted in a different optimal spin-up period.
**We agree. Accordingly, we introduced 'for our purposes'.**

29. Page 23, line 20: What are slow-reacting features?
**We changed the line to 'slow-reacting variables like soil moisture'. Note that we come back here to something we introduced in Section 2.3.**

30. Page 23, line 20: Again 'good results' depends on what you are trying to achieve. Small differences in the ensemble will occur if the cyclones are already present in the outermost model domain. Is that the point?
**We agree that we should be more specific here and expand on the suggestion of the reviewer in the revised manuscript.**

31. Page 24, line 7: I do not think you can conclude that uncertainty increases gradually when going back in time using 3 case studies only.
**In fact, we do not think that we generalize here, given the context of the paragraph, and the first word of the sentence being 'This', that is the uncertainty in our cases. Note however the added sentence in the conclusions: 'Although going back far in time, we have only analyzed a very small number of events – many more cases would be needed to reach robust recommendations on how to configure a model for Vb cases.'**

32. Page 24, line 9: Similarly, concluding that dynamical downscaling is less accurate going back in time is difficult using 3 case studies only. There are many other factors that would increase the uncertainty for specific case studies.
**See the reply to comment 31.**

33. Page 24, lines 14-23: This is an excellent summary and it would be nice to see a more in-depth analysis in the main body of the text to support these conclusions.
**We tried to corroborate these conclusions with the extended analyses of the 'cyclonic moisture flux', including the new Figure 11 and an additional clause in the Abstract.**

34. Page 24, line 26: How do you conclude that the 20CR tracks are not 'realistically located' for the 1876 case? Are you stating that 20CR produces unrealistic tracks, or simply that the uncertainty in the position of the track is large for this case potentially because it is a complex situation?
**Realistic is indeed the wrong term here; we thank the reviewer for pointing this out. We rephrased it to 'variability becomes very large', and we add that some cyclone tracks do not follow the classical Vb path anymore. Note also the extended explanations of uncertainty.**

**Further changes made**
- **Additional references: Hofstätter and Blöschl, 2019; Zbinden, 2005 (Annals of MeteoSwiss); Cioni and Hohenegger, 2019; Coppola et al., 2018; Compo et al, 2015; Joliffe and Stephenson, 2012; Wernli et al., 2008.**
- **Figure A2 redrawn with the maximum- and minimum-precipitation members. This makes more sense since we focus on these in the text.**

---

## Author Comment (AC3) · 27 Sep 2019

Review of nhess-2019-174:

"Simulations of the 2005, 1910 and 1876 Vb cyclones over the Alps – Sensitivity to model physics and cyclonic moisture flux"
by
Peter Stucki, Paul Froidevaux, Marcelo Zamuriano, Francesco Alessandro Isotta, Martina Messmer, Andrey Martynov

Recommendation: major revisions

The authors present three cases of extreme precipitation along the Alps associated with the classical synoptic situation Vb. They utilize downscaling of different reanalysis products for their analysis, where they employ WRF in a nested method to go down to 3 km resolution in the innermost domain. The authors test several sensitivities of their results with respect to the simulation setup, addressing both numerical as well as physical changes. They find that the lead time is among the most crucial parameters. While I find the manuscript well written, I struggle to see a clear motivation and conclusion for the study. The motivation is probably also somewhat difficult, because the manuscript tries to address many different questions at the same time: (1) What is the best downscaling setup for the cases in question. (2) What lead to the extreme precipitation. (3) Description of
the three individual cases and comparison. The treatment of all these topics makes the paper sometimes difficult to follow. Regarding the conclusions, similar arguments apply and the authors sometimes appear to state opinions/speculations that are not necessarily solidly grounded in the material they presented. However, after responding to some of my major concerns, I believe this manuscript is acceptable for publication in NHESS.

**We thank the reviewer for the comments and helpful, concrete suggestions for many aspects and important details. Many of the comments made us rethink our argumentation, and we found ways to better explain a topic or a reasoning based on this review. At times, this needed a new paragraph, at other times a couple of extra words were sufficient. With all these efforts taken, we think that the reviewer contributed largely to an enhanced, more specific and clearer manuscript. We hope that he / she will also more easily find our general line of thought in the revised manuscript - from first finding a setup, and then applying it to assess precipitation and moisture flux in the three cases.**

General Comments:

In the title, I am not sure what the authors really mean with the "cyclonic moisture flux". Is it a moisture flux going in a cyclonic direction or the moisture flux associated with a cyclone? Given this ambiguity, I encourage the authors to further clarify this aspect in their title as well as throughout the manuscript to help the reader making a clear link.

**Indeed, the comment has made us re-think about what is meant by 'cyclonic' and 'cyclonic moisture flux'. We come to the conclusion that both aspects (cyclonic rotation and association to a cyclone) are relevant in our case, although we mainly refer to it as 'anti-clockwise'. To be clearer, we've added an explicit description of the specific dynamic mechanism of what we call a 'cyclonic moisture flux'. This can be found in the Introduction section.**

**We also see that the term 'cyclonic moisture flux' is indeed not commonly used in publication titles or abstracts. However, we have not found a less specific, but still correct and concise term that would substitute it in the title. For this, and because it is also a relatively important term in our article, we would like to keep it there. We think that most readers may have some notion of what is meant when they read the title, and will be ready to read up about it in the introduction.**

Given that there are only three cases, and the fact that a majority of the reasoning for the downscaling is based on the most recent case, the authors should be more cautious about general statements on downscaling procedures, as the results are highly sensitive to the case(s) at hand. While the technicalities that were overcome by the group are certainly impressive, it is still not clear to me how generic these results can be treated. In order to make a more general claim about the downscaling for Vb situation, one would need to explore many more cases to arrive at a firm conclusion. The authors should thus make it clear that this study can at most give an indication what one might need to test in order to arrive at a more general conclusion.

**According to the reviewers' suggestion, we have tried to avoid the impression of making general conclusions throughout the manuscript. This is mostly done by adding some words like 'for our purposes', 'in our case', 'our experiments', etc. We have also added a clear statement in the conclusions: 'Although going back far in time, we have only analyzed a very small number of events – many more cases would be needed to reach robust recommendations on how to configure a model for Vb cases. Nevertheless, we have demonstrated that one can achieve a relatively best configuration for the desired application with a well-thought, logical series of experiments.'**

What made the authors pick a 10-day spin-up time? It seems excessively long for the investigation of such a regional and meso-scale influenced precipitation event. At the end, the authors arrive at a 1-day spin-up time anyway, but the vastness of the parameter space is not sufficiently motivated, similar to some of the other sensitivity tests.
The list of sensitivities is extremely exhaustive, ranging from resolution to resolution ratios over spin-up time to parameterizations and model domains. The enormous parameter space is rather difficult to grasp and all results will primarily be in relation to

the 2005 case, with general deductions being rather limited due to the specifics of the case. In general, it would aid the reader if the authors more clearly state their working hypotheses as well as the reasoning for their choices and expectations. This will make it more straight forward to follow the ensuing arguments.

**Based on these comments, see also the specific comment below, we now state more clearly in Section 3.1 what (i) our initial expectation was (10-day spin-up allows soil moisture and other variables to reach a partial equilibrium), (ii) that unsatisfactory first results led to experiments with reduced spin-up, and (iii) when the best setup was found, checking its robustness by testing additional changes of the model configuration. This reasoning is now also better underpinned with references.**

**Also, we state more clearly that '**The goal of these last experiments is not to achieve a thorough sensitivity assessment for each tuning option, but to make sure that we have not chosen a sub-optimal setup. This is checked by further modifying a number of configurations of the WRF model which may have an influence on the simulation performance according to literature and from our experience'**. In addition, we have introduced more literature and some short explanations of why we check against another (technical) configuration. In the Abstract, we focus on the result concerning the initialization time only because it is best documented in the manuscript.**

The authors often refer to reproducing "correct" precipitation amounts. What is meant by correct? Presumably compared to observations, though the authors list several observations that are used. In addition, all of these "observations" also rely on some sort of downscaling and gridding, as data voids need to be filled. The authors, however, do not provide a detailed analysis of the representativeness of these observations. They refer to other studies that addressed these to some extent, but given the specifics of the case studies, the authors should also comment on the validity of the observations before comparing the model simulations to the data in order to claim "correctness".

**We think that the term 'correct' describes well what we are striving for in the experiments, and therefore prefer to keep it in the manuscript. To be more specific about what we mean by correct, we have tried to include a reference where possible throughout the manuscript. For instance, we insert 'that would ideally produce', 'i.e., in the control area', 'compared to CombiPrecip'.**

**We are aware that the interpolations (as well as the original measurements) are affected by uncertainties and errors. The uncertainties and errors were analyzed in detail as described in the publications mentioned in the manuscript as well as in many applications (e.g. at MeteoSwiss). In-depth knowledge about the strengths and deficiencies of each datasets allows MeteoSwiss to make comparisons always considering the performance of the datasets used as reference. A detailed description of the representativeness of the observation for each case would replicate already published results and go beyond the scope of the present study. In section 2.1, we add a comment about our awareness about the uncertainties and errors of our datasets (and the consequential appropriate use).**

**The fact that more than one dataset is used is due to their availability (e.g. CombiPrecip is not available for the 1910 case) and the differentiated comparison with grid data and time series.**

For the validation of, for example, precipitation, it has proved useful to use feature-based detections that consider location, shape, and timing. Why have the authors not considered more such verification tools for the study at hand? It appears the method referred to as EMD is in fact such a measure, though it appears confusing why the authors use a visual inspection for a quantitative comparison. The reasoning for the choices and omission of other tools should be clearly motivated.

**The same as the reviewer, we see EMD as a kind of feature-based measure. In fact, we tested other measures, mostly using the spatialVx package for R (https://ral.ucar.edu/projects/icp/references.html), including SAL by Wernli and Schwierz, 2006. Advantages of the EMS were in our view: smaller number of subjective decisions (thresholds, smoothing options etc.), one instead of three measures, no failures of actual feature detection, a measure for the relative distribution of precipitation totals. As the reviewer suggests, we have summarized these reasons in the text.**

**Regarding the 'eyeball inspection', it seems to be a common and valid option and complement to machine techniques for pattern recognition, see**
**https://www.cawcr.gov.au/projects/verification/#Standard_verification_methods**
**https://www.swpc.noaa.gov/sites/default/files/images/u30/Spatial%20Forecast%20Verification.pdf**
**We do agree on the point that more reasons are required to justify the selection of measures and scores, though. Therefore, we have introduced some short justifications in Section 3.2.**

For the philosophical concluding paragraph on page 25 not much hard evidence has been provided in the manuscript for the claims put forward. It thus reads more like a written piece of opinion than a well and quantitatively justified conclusions.

**We suppose that the reviewer alludes to the use of 'propositions of possible weather'. We see this rather as an intuitive and informative description of what the ensemble provides in the end, and think this could be appropriate in a final paragraph. Apart from that, we do not see any other philosophical conclusions on page 25. We rather think that we describe options that have been tested already, or paths that could be taken to balance computational costs and meaningfulness of downscaling an ensemble.**

Specific Comments:

The page (P) and line (L) numbers refer to the ones in the manuscript.
P1 L23: "to the cyclonic" and see comment above about the ambiguity of "cyclonic moisture flux".
**Please refer to our reply to the general comments above.**

P1 L28: "accurate directions" with respect to what? What is the reference?
**We agree that 'accurate' is too vague here and have changed it to 'northerly'.**

P8 L1: The precipitation data is interpolated. Can the authors please clarify if the interpolation was carried out in such a way that the total precipitation was unaffected by

the interpolation? Depending on the kind of interpolation, the results for the totals can deviate.

**Please refer to our reply to the general comments above (fourth comment). We added some sentences in Section 2.1 to illustrate our awareness about the interpolated datasets.**

P11 L12: The authors speculate on the differences between ERA-Interim and 20CR in terms of moisture distribution, though the authors could provide direct evidence for their claim by investigating differences between ERA-Interim and 20CR fields for the case at hand in more detail.

**Following this comment, we have added some more information about a short investigation of specific moisture at 1000 hPa.**

P11 L22: Why did the authors chose to identify and track cyclones using geopotential at 500 hPa? This seems rather unconventional and needs further motivation.

**We agree that it is a good idea to extend the explanation of our choices. Concretely, we introduce the tracking paragraph with: '**For the analyses, we use both sea level pressure (SLP) and mid-tropospheric pressure fields. SLP fields inform about the quality of the assimilation process in 20CR, and the isobaric pressure fields (at 500 hPa here) tell about the derivation of upper-air variables from the SLP information in 20CR. Combining SLP and isobaric levels has been found useful for cyclone tracking (Hofstätter and Blöschl, 2019).**'**

Fig. 4: The cyclone tracks for ERA-Interim look very edgy. In order to compare them better to the other plots, a grid that is not fixed to the grid spacing of the data could be beneficial, which is most often done in other cyclone track algorithms, see also comment above about cyclone track determination in this manuscript.

**We agree on the 'edginess', but would prefer to leave the visualization as is in Figure 4d. In our view, it reflects the nature (the coarse resolution) of the underlying data; a smoothed version would rather mask it. In addition, we see the ERAI line as a rather complementary information in the plot. Note also that for the same reason, we did not interpolate the grid points in Fig. g – i, and included the original grid points in the new Figure 11b, d and f. For other variables however, we see that smoothing is necessary, e.g. for cyclone tracks or SLP.**

Fig. 7, 8, and 9: I find these figures not very legible. Maybe this is due to the downgrading of the figure quality for the review process, but otherwise the readability of the information of these figures needs to be significantly improved. In particular the arrows are not very visible.

**We fully agree on that, and all reviewers are of the same opinion. Unfortunately, the quality loss was introduced by the conversion to PNG format. This should mostly be enhanced with the current PNG, and the PDF for print is fine as well, in our view. Note also that the color key had been missing.**

P22 L21: The authors should explain how PV is produced in the downscaling process, as

this appears to be crucial in their arguments.

**In accordance to a similar comment by Reviewer 2, we have rephrased the discussion without PV streamers. The aim of the paragraph is to mention the underlying physical processes and interaction between the Alpine orography and upper-level troughs. These physical processes can be referred to without introducing the notion of PV, which is indeed not introduced in our study. In fact, this has allowed to focus more on explaining the untypical results from the 1876 case. The new text reads as follows:**
"This behavior of the model can be explained by the interaction of the Alpine orography with atmospheric circulation. Indeed, Vb cyclone trajectories are typically initiated by deepening upper-level troughs, which finally cut off from the westerly flow when passing over the Alps (e.g. Awan and Formayer, 2017). The interaction of upper-level troughs with the Alpine orography have been described in detail (Buzzi and Tibaldi 1978; Aebischer and Schär 1998; Kljun et al. 2001); the underlying processes include flow splitting and lee cyclogenesis, with further amplifications of the cyclone formation by frontal retardation and latent heat release due to orographic lifting. The combination of these processes implies that the cyclones are formed on the lee of the right side of the Alps, typically over the Ligurian Sea. In 20CR, however, the Alpine orography is very coarse, smoothed and reaches only about 1000 m a.s.l. (cf. Stucki et al., 2012). Hence, the influence of the Alps on the large-scale flow is limited in 20CR. Given also that the 1876 case is least confined by pressure observations, this allows untypical cyclone tracks in many 20CR members. Once accounting for a more and more realistic orography throughout the downscaling steps with WRF, the high-resolution runs may thus end up in a compromise simulation - driven both by the WRF model physics and by the 20CR input flow. In other terms, the large-scale flow forced from 20CR might not be compatible with the orography of the high-resolution domains."

P23 L24: How can the authors conclude that "nudging smaller domains can still be beneficial"? Has any evidence been provided in this study to support such a claim?
**We agree that this could be understood as a claim. Therefore, we rephrased the sentence to: '**Although we find no relevant enhancements from nudging in smaller domains in our test experiments, nudging smaller domains could still be beneficial for other specific studies.**'**

P24 L1: The authors should be more specific what they are referring to with "traditional spatial verification scores".
**Regarding traditional (or standard) methods, we referred to https://www.wmo.int/pages/prog/arep/wwrp/new/jwgfvr.html, among others. However, we see that the term might be confusing, and substituted it by 'more common', and we name our measures again.**

**Further changes made**
- **Additional references: Hofstätter and Blöschl, 2019; Zbinden, 2005 (Annals of MeteoSwiss); Cioni and Hohenegger, 2019; Coppola et al., 2018; Compo et al, 2015; Joliffe and Stephenson, 2012; Wernli et al., 2008.**
- **Figure A2 redrawn with the maximum- and minimum-precipitation members. This makes more sense since we focus on these in the text.**

---

## Author Response (AR1)

**Author's response**

To nhess-2019-174 Submitted on 31 May 2019

Simulations of the 2005, 1910 and 1876 Vb cyclones over the Alps – Sensitivity to model physics and cyclonic moisture flux

Peter Stucki, Paul Froidevaux, Marcelo Zamuriano, Francesco Alessandro Isotta, Martina Messmer, and Andrey Martynov

**Contents**

- List of relevant changes made in the manuscript
- Marked-up manuscript version
- Response to Reviewer 1
- Response to Reviewer 2
- Response to Reviewer 3

30-09-2019, Peter Stucki

**List of relevant changes made in the manuscript**

(page number in brackets refers to the pdf with markups below)

**Abstract**

- Contribution of (embedded) convection added (p.1)

**Introduction**

- Enhanced definition of 'cyclonic' and 'cyclonic moisture flux' (p. 2)

**Data and Methods**

- Extended description of RrecabsD gridded reconstruction of precipitation (p. 4)
- Number of stations in Switzerland that are assimilated into 20CR for 1876, 1910, 2005 (p. 4)

**Downscaling of 2005 case / Evaluation of setup**

- Rationale for extended initialization period for WRF simulation (p. 5)
- Rephrasing of results from Fig. 2 wrt. CombiPrecip gridded dataset (p. 6)
- Reasoning for additional experiments after evaluation of the standard WRF setup (p. 6, 7)
- Extended description of WRF configurations (p. 7)
- Extended description of evaluation measures (MAE, EMD, Theil-Sen slopes; p. 9)
- Extended discussion of setup evaluation (p. 10)

Analyses and simulations of three cases

- Extended description of SLP and isobaric pressure fields for cyclone tracking (p. 10)
- Short description of Wernli and Schwierz 2006 cyclone tracking method (p.11)
- Extended analyses of Fig. 4, enhanced caption Fig. 4 (p.14.15)
- Added panels to Fig. 5 with 9-km domain and convective contribution (p. 15)
- Extended analyses of Fig. 5 (p. 15, 16)
- Extended analyses of Figs. 7-9, color key introduced (p.18, 19, 20)
- Rephrased interpretation of analyses Figs. 7 -10 (p. 22)
- New analyses of new Fig. 11 concerning innermost domains (p.22, 23)

**Conclusions**

- Extended explanation of 'small-scale adaptation' and influence of large-scale flow (p. 24)
- Enhanced conclusion on increasing variability in the ensemble when going back in time (p. 25)
- Inclusion of single-member analyses in new Fig. 11 (p. 25)

**Appendix**

- Fig. A2 redrawn with maximum and minimum members

**References**

- A number of additional references given

**Figures and Tables**

- Fig. 1 redrawn (no edges cut)
- Fig. 2 includes member number
- Fig. 4 redrawn (panels g- i)
- Fig. 5 redrawn (new panels of 9-km domain, WRF-ERA-Interim added)
- Figs. 7-9 redrawn with color key, enhanced raster image quality
- New Fig. 11
- Fig. A2 redrawn with max. and min. member

**Simulations of the 2005, 1910 and 1876 Vb cyclones over the Alps – Sensitivity to model physics and cyclonic moisture flux**

Peter Stucki1,2, Paul Froidevaux2,4, Marcelo Zamuriano1,2, Francesco Alessandro Isotta4, Martina Messmer1,3,5, Andrey Martynov1,2

1Oeschger Centre for Climate Change Research, University of Bern, Bern, 3012, Switzerland
 2Institute of Geography, University of Bern, Bern, 3012, Switzerland
 3Climate and Environmental Physics, Physics Institute, University of Bern, Bern, 3012, Switzerland
 4Federal Office for Meteorology and Climatology MeteoSwiss, Zurich-Airport, 8058, Switzerland
 5now at: School of Earth Sciences, The University of Melbourne, Melbourne, Victoria, Australia

Correspondence to: Peter Stucki (peter.stucki@giub.unibe.ch)

Abstract. In June 1876, June 1910 and August 2005, northern Switzerland was severely impacted by heavy precipitation and extreme floods. Although occurring in three different centuries, all three events featured very similar precipitation patterns and an extra-tropical storm following a cyclonic, so-called Vb trajectory around the Alps. Going back in time from the recent to the historical cases, we explore the potential of dynamical downscaling a global reanalysis product from a grid size of 220 km to 3 km. We use the full, 56-member ensemble provided in the reanalysis and a regional weather model to investigate sensitivities of the simulated precipitation amounts to a

- 20 set of differing model configurations. The best-performing model configuration in the evaluation, featuring a 1day initialization period, is then applied to assess the sensitivity of simulated precipitation totals to cyclonic moisture flux along the downscaling steps. The analyses show that cyclone fields and tracks are well defined in the reanalysis ensemble for the 2005 and 1910 cases, while deviations increase for the 1876 case. In the downscaled ensemble, the accuracy of simulated precipitation totals is closely linked to the exact trajectory and stalling position
- 25 of the cyclone, with slight shifts producing erroneous precipitation, e.g., due to a break-up of the vortex if simulated too close to the Alpine topography. Simulated precipitation totals only reach the observed ones if the simulation includes continuous moisture fluxes of >200 kg m-1 s-1 from northerly directions, and high contributions of (embedded) convection. Misplacements of the vortex and concurrent uncertainties in simulating convection, in particular for the 1876 case, point to limitations of downscaling from coarse input for such complex weather situations and for the more distant past. On the upside, single (contrasting) members of the historical cases are well

capable of illustrating variants of Vb cyclone dynamics and features along the downscaling steps,

**1** Introduction**

Floods are among the most damaging natural hazards worldwide (Bevere et al., 2018); they affect more people than any other natural hazard (CRED 2019). The costliest flood event in Switzerland of the last decades occurred in

35 2005 (Hilker et al. 2009); it caused fatalities and led to heavily damaged infrastructure (Bezzola et al. 2008). This event was well documented and subsequently, a range of publications analyzed the flood-inducing meteorological conditions (e.g., Frei 2005; Beniston 2006; MeteoSwiss 2006; Bezzola and Hegg 2007; Zängl 2007a; Bezzola and Hegg 2008; Hohenegger et al. 2008; Jaun et al. 2008; Langhans et al. 2011; Stucki et al. 2012; Messmer et al. 2017).

[revised manuscript text omitted]

---

## Author Response (AR2)

**Author's response Round 2**

To nhess-2019-174    Submitted on 31 May 2019

Simulations of the 2005, 1910 and 1876 Vb cyclones over the Alps – Sensitivity to model physics and cyclonic moisture flux

Peter Stucki, Paul Froidevaux, Marcelo Zamuriano, Francesco Alessandro Isotta, Martina Messmer, and Andrey Martynov

**Contents**

- List of relevant changes made in the manuscript (with regards to revisions in the first round)
- Comments by Referee #4 and point-by-point response
- Marked-up manuscript version

15-11-2019, Peter Stucki

**List of relevant changes made in the manuscript**

(page number in brackets refers to the pdf with markups below)

Abstract

- Explanation of cyclone fields and tracks in parentheses (p.1)

Introduction

- Enhanced reasoning on using a full ensemble (p. 3)

Data and Models

- Description of final model instead of original model (p. 5)

Setup and Evaluation

- Section titles changed to 3.1 Setup … and 3.2 Evaluation (p. 5 and 7)
- Extended explanation of dimensions for the evaluation (spin-up and tuning options; p 5)
- Fig. 2 moved to Appendix (new Fig. A2), description of Fig. 2 cut to basic information (p. 7)
- Enhanced reader guidance regarding evaluation of spin-up and tuning options (p.8 )

Analyses and simulations of three cases

- Enhanced justification for using surface and tropospheric levels for cyclone tracking (p. 9)
- Introduction of the three analyses in section 4.2 (p. 13)
- Changed explanation for errors in initialization period (p. 15)

Appendix

- Fig. A2 added

References

- One reference deleted

Figures and Tables

- Fig. 2  relocated to Appendix
- Fig. Numbers adapted accordingly

Reviewer comment on "Simulations of the 2005, 1910 and 1876 Vb cyclones over the Alps – Sensitivity to model physics and cyclonic moisture flux" by Peter Stucki et al.

Report #2
Submitted on 18 Oct 2019
Anonymous Referee #4

Suggestions for revision

This manuscript describes a downscaling exercise of three major Alpine flooding events in three different centuries. The main findings of the study, in my opinion, are (i) the major role of the relationship between cyclone location and moisture fluxes towards the Alpine slope in determining the regional precipitation patterns and (ii) the insight that a pre-selection of ensemble members according to these large-scale forcing factors (cyclone position, moisture flux) may be useful prior to dynamical downscaling. While point (i) has already been made in previous studies, the new aspect here is that its relevance is demonstrated consistently through a downscaling chain.
In this revised version, the authors have put a lot of effort in incorporated comments and suggestions by three reviewers. Note that I have not been part of the first review round and have thus looked at the manuscript for the first time after these revisions. Possibly also related to this later involvement, my opinion on the manuscript is somewhat different from the previous assessments. While I think that the findings summarized above are clearly important and worth publishing, and the paper is technically and methodologically sound, the way the manuscript is put together, in my view, is not very appropriate. My main criticism is related to the fact that the paper is full of technical details and it is very difficult to extract the main messages and conclusions. As it is often the case, the number of details has even been increased by incorporating the comments from the first review round. I'd be really afraid that many readers will give up during the first technical part before being able to appreciate the interesting results described towards the end of the paper. I'd thus invite the authors to take a step back and think again which of their results provide useful insights for other researchers in the fields and which steps, on the other hand, may have been technically important during the preparation of this study, but do not necessarily have to be described in full detail in the paper.

**We thank the reviewer for the suggestions for revision. Of course, we like to read that the article is thought to be technically and methodologically sound, and we are grateful for a number of very helpful specific comments.**
**We also agree that the number of details has been increased by adopting request from the previous reviews. As a consequence, it is obviously quite difficult to balance the manuscript such as to satisfy old and new requests, a delicate situation for us.**
**In the end, we have implemented suggestions by the referee #4 where we see no or only minor conflicts with requests from the previous round and by the second referee of this round. Otherwise, we give reasons where we do not follow the suggestions by referee #4. Foremost, these are requests to reduce the information in the figure panels again, and to discard or move an additional analysis. We give our reasoning in the respective replies below.**
**Nevertheless, we have also changed substantial parts of the manuscrit to restructure the text, avoid redundant information and reduce the number of details. This is particularly the case for the rather technical aspects in the data and methods chapters. Concretely, we have moved**

**one figure to the Appendix and cut the respective paragraph accordingly. Then, we give the main information about the selected setup in the data chapter. This means that the full Evaluation chapter can be skipped by readers who are not interested in these rather technical aspects. Furthermore, we have restructured Chapter 3 to avoid a chronological order and get rid of some redundant information, and we have enhanced reader guidance in Section 4.2. We are confident that the earlier referees can agree on these changes because in sum, we think that the incorporated suggestions clearly enhance the quality and readability of the article.**

Here a few more concrete comments that might be helpful in this respect:

- The entire section 3 reads more like a work report than a scientific paper (the structure, in short, is "we started with a specific approach, than tried something else that turned out to be better, and finally tested a few more sensitivities"). For a paper, such a "chronological" order is often not the best choice, and I'm really not sure what a reader should learn from this lengthy description (as you only analyzed a few cases, your specific results, e.g., on the choice of parameterizations can hardly be generalized anyway). I'd suggest to start with the description of the main setup that was used as a basis for the process analysis in section 4 (lead time of 1 day) and afterwards briefly (!) summarize the sensitivities w.r.t. lead time and parameterizations (removing most of the details, e.g., provided in tables 1 and 2, or moving them to the appendix).

**We can see moist of the points made in this comment. Nevertheless, it expresses also views that produce a conflict to requests made by reviewers in the first round. The main controversy we see is that we were specifically asked to better balance the model evaluation part by extending the dynamical analyses of innermost domains. Now, deleting or moving large parts of the evaluation to the Appendix would introduce another imbalance. Furthermore, we were specifically asked to give more technical details about the chosen methods and tested options. For these reasons, we think we cannot cut down this part too heavily.**

**However, we have found a way to avoid the 'work report' approach, which also allows to shorten the text, reduce redundant information, and allows readers who are not interested in the evaluation to skip this section completely. As suggested, we now describe the final model configuration in the Data and Models section. In the Evaluation section, we then describe the two dimensions of evaluation (spin-up and tuning options) and introduce the evaluation tools. As a consequence, also the discussion of the evaluation has now become more consice.**
**The new approach even allows to move a figure to the Appendix (Fig. 2 old is now Fig. A2), as it has some redundance with Fig 5 old (Fig. 4 new). On the downside, we can only explain the subset of ensemble members very briefly. The according text reads now as follows:**
**"To save computational costs, the experiments are done with a subset of ten members that cover the full range of precipitation variability from a first-guess setup (not shown). This original setup (in fact, the full ensemble of the 10-day initialization setup) resulted in a general underestimation of the accumulated precipitation in the control area compared to CombiPrecip, the reference dataset (Figure A2 in the Appendix)."**

- Much related to the previous point: I'm really not surprised by the fact that shorter lead times lead to better results. This first-order relevance of the initial conditions is very well known from weather prediction. Compared to this, soil moisture conditions are clearly of secondary importance, and in

addition it may take much longer than 10 days for the soil moisture to equilibrate if it is not properly initialized. In this sense, the author's first approach has been somewhat naive, and I do not think that it is necessary to dedicate a lot of space to describing the fact that shorter lead time yield better results.

**We think that with the new structure of Section 3 (see the reply above), we can avoid the impression of having used a naïve approach, and rather emphasize the fact that we checked in every direction to make sure we have not selected a suboptimal model configuration.**

- The description of sensitivities to the parameterization schemes is spread over the manuscript (e.g., end of section 2, section 3.2), which should be avoided.

**Following the suggestion made in the first concrete comment above, we describe the final selection in Section 2, and sensitivities are assessed in Section 3. This allows the reader to skip Section 3 without missing important information about the model used.**

- The potential pre-selection of ensemble members based on their representation of the relevant large-scale conditions (my point (ii) in the first paragraph) is only mentioned at the very end of the paper, where it might be overseen. I think that this is an important general conclusion from the paper that should be highlighted more specifically. I would add a note on this at the end of the abstract and also formulate a corresponding research question/hypothesis in the introduction section to prepare the reader for this finding.

**In the first submission, we had a sentence about potential pre-selection in the abstract, and hints to it in the introduction.**
**However, we replaced it because of reservations by one reviewer from the first round, regarding the quantitative evidence for this conclusion. We agree in that we might need more specific analyses (e.g., have the downscaled 'good' members already been 'good' in 20CR?). The way we present it now is rather "that we describe options that have been tested already, or paths that could be taken to balance computational costs and meaningfulness of downscaling an ensemble.", as we replied to the reviewer.**
**This means that we would like not to emphasize this conclusion too much, although we agree in that it might provide some guidance for other downscaling experiments.**

- I do not find it very helpful to add the specific case study at the end of section 4. This again shifts the focus to the details ("On 15 June 1910, 8 UTC…") and away from the more general conclusions. If you really want to keep this, I would move it to the beginning of the results section to set the scene for the following, more quantitative analysis.

**We agree with the referee that this final part introduces more details, which may not be of interest for some readers. However, we do not think we can discard the analyses; for two reasons. The first is that we were explicitly asked in the first reviewing round to extend the dynamical analyses of the two innermost domains, and that the second reviewer in the second round has approved these changes, obviously. The second reason is that with this analysis, we can show that the single member proposes very plausible weather situations for a 100-year old case, with a number of striking analogs to the 2005 case.**

**We have also explored ways to move the analysis to the beginning of the results, to no avail. This is because it seems very odd to explain why we start with a single maximum-precipitation member, and because this analysis is referring to the IVT / moisture flux analyses, which would require a 'spoiler' paragraph. We would therefore prefer to stick with the current series of paragraphs, going from the full ensemble to contrasting members, then from large-scale to small-scale along the downscaling steps.**

**However, we clearly see that the amount of provided details in this section may be tiring. To provide a better reader guidance, i.e. such that they will be free to read all or selected analyses only, we now explain our line of thought for the analyses at the beginning of the section. It reads as follows:**

**In the first place, we analyze the variability of precipitation totals in the downscaled ensemble for all three cases, with a special focus on the contribution of convection. Secondly, we address flow features along the downscaling steps which explain contrasting (very high or very little) precipitation totals in the simulations. To conclude the analyses, we use an exemplary member of the 1910 case to illustrate how a specific pattern of cyclonic flow translates into characteristic near-surface weather during heavy-precipitation Vb events over the Central Alps.**

Specific comments:

Other minor/technical comments (page and line numbers refer to the version with marked changes in the response letter):

P 1, l 22-23: The sentence "The analyses show…" is difficult to understand at this point because it is not generally clear what "cyclone fields" are. I would also make clear that "deviations" in the second part refer to deviations between ensemble members (and not with respect to some ground truth).

**Indeed, the term cyclone fields is only introduced late, that is in Chapter 4. Hence, we introduced a parenthesis (closed pressure contours) to explain what we mean. To be consistent, we also explain cyclone track (minimum pressure trajectories).**

**We also extended to 'deviations from the ensemble mean'.**

**Some other text is removed in the Abstract to reach the maximum 300 words allowed.**

P 2, l 6: "For this reason" (add reason); bracket missing

**The suggestion is adopted. We thank the reviewer for pointing out the typo.**

P 2 , l 14, "transported" or "advected" instead of "led"

**We have adopted the suggestion with 'transported'.**

P 2, l 36 – P 3, l 5: This paragraph is rather technical for an introduction section, and it left me with more questions than answers (which approach did you use? which is more suitable?)

**We have replaced a more technical sentence by a sentence that explains better our approach with the full ensemble. Because we have taken out the aspect of pre-selecting members, we have also dropped a respective sentence here.**

P 3, l 12: "losing control over" is too unspecific; the information from the initial conditions is lost

**The respective clause is changed to 'although the information from the initial conditions is lost with advancing time in the simulation'.**

P 3, l 14-17: I'd avoid the term "cloud-resolving" (although it has been previously used in the literature), as a parameterization of cloud microphysics is still required. In this context, also "explicit production of precipitation" is imprecise.
**We would like to keep the terms as they are. We agree that 'cloud-resolving' might not be the most concise term. However, it is widely accepted in the community, and alternatives like 'storm-resolving', 'convection-permitting' have not gathered more acceptance, as the audience concluded in a discussion at the recent Latsis symposium at ETH Zürich. Similarly, the term 'explicit' might not be very precise, but we think that informed readers will immediately know what is meant**.

P 4, l 20: I don't understand the term "1st-step deviations"
**We have discarded this technical detail. It alludes to the fact that in the production process of 20CR, the ensemble members are set up as 'deviations'**.

P 5, l 16: "parameterizing" instead of "resolving"
**The suggestion is adopted.**

Section 4.1: You put some effort in identifying cyclones on different vertical levels, but it is not entirely clear to me what you learn from this in the end. Can you formulate a conclusion that could not have been obtained from detecting cyclones on one level only?
**We think that this may be a difficult question to answer from our analyses.**
**Including two levels is rather a safety measure to avoid pitfalls that may arise from analyzing the surface pressure fields alone. To address the obviously lacking explanation, we have included two references and summarized their rationales for using surface and isobaric levels. The text reads now as follows:**
**Combining SLP and isobaric levels has been found useful for cyclone tracking (Hofstätter and Blöschl, 2019; Hofstätter et al., 2016): While SLP tracks and fields accurately mark the "footprint" of a cyclone, surface pressure patterns are also modulated by (model) orography and boundary layers and can therefore behave more short-lived and erratic in space. In this respect, pressure patterns at mid-tropospheric levels are smoother and thus more robust over space and time.**

P 16, l 9-12: I cannot follow this. Doesn't the interpolation from the coarse input data apply a height correction? How do warm temperatures "produce" water vapor?
**An explanation for this behavior in the initialization period was asked for by a previous reviewer. We have tried to be more clear by rephrasing to:**
**One explanation might be found in the coarser interpolation from 20CR input data (from 220-km grids compared to 80-km in ERA-Interim), which results in too high temperatures over the Alps for this specific case. In turn, this may lead to enhanced convection over this area and during the first hours of the 20CR-WRF simulation.**

P 24, l 21: "parameterizations are turned off": no, only the convection scheme
**The suggestion by the reviewer is adopted.**

P 25, l 18-20: It is impossible to say if the "patterns and dynamics" are reproduced by some ensemble members also for the 1876 case, as there is no ground truth for comparison. From the ensemble spread, you can only learn that the uncertainties are larger because of missing observational constraints.

**There might be a misunderstanding here. In the referred lines, we use 'reproduced' for the 2005 and 1910 cases. This is because we have a quite large number of observations (plus radar for 2005) and a high-quality reconstruction (RrecabsD) as a reference (see Sect. 2 and Fig. A1). Because of comparably few observations and stations that went into RrecabsD, we refrain from using 'reproduced' for the 1876 case, but only state that the "variability of the cyclone fields and tracks becomes very large in the 20CR ensemble for the 1876 case". This is similar to what the reviewer says in the last clause of the comment above.**

Figures: Most figures contain a lot of different information, which reduces their accessibility. Please check all figures once again and remove the content that is not absolutely necessary to demonstrate the main findings (for instance, showing results from an ERA-Interim driven experiment in Fig. 5 is again very much on the technical side; I'm also not sure why it should be helpful to mention this experiment in the main text on page 16).

**This is a delicate issue for us, because the suggestion is in conflict with the requests by the earlier reviewers, and because we invested quite some time in redrawing the figures accordingly. For this, we would really like to leave the figures as they are. However, we have moved one figure to the Appendix, which reduces the amount of complexity to digest in the main article. More detailed explanations regarding each figure are as follows:**

**Figs 1, 3 and 6, as well as A1, A2 and A3 are not very complicated, addressing one or two parameters only. Figure A2 has been Fig. 2 in the old manuscript.**

**Fig. 3 has been redrawn to make the essential messages more clear, addressing concerns raised by the earlier reviewers. We discussed moving one or two rows to the Appendix, but then preferred to keep everything together.**

**Fig. 4 has indeed been expanded. However, this was done because one reviewer explicitly wished a more detailed analysis of the innermost domains. To compensate, we deleted information from previous versions. The comparison to ERA-Interim was necessary, among others, to address an issue raised by one reviewer about the initialization period.**

**Figs. 6-8 consist of many panels, indeed. However, they address just one parameter in the same way, and we already discarded the outer- and the innermost domains.**

**Fig. 9 is indeed a quite complicated figure. After briefly discussing the options, we decided to leave it as is, because similar figures have been published in Froidevaux and Martius, 2016, for instance, and because shown information cannot easily be omitted without losing the message at all.**

**Fig. 10 is again a busy panel, of course. However, it has the same three variables (pressure, precipitation, wind) in each panel displayed in the same manner. We had checked already to reduce the information further while preparing this new figure, particularly wind vectors and isobar increments, but we would have lost too much information.**

[revised manuscript text omitted]